# MADE: Exploration via Maximizing Deviation from Explored Regions

**Tianjun Zhang**[*1]    **Paria Rashidinejad**[*1]    **Jiantao Jiao**[1]    **Yuandong Tian**[2]

**Joseph E. Gonzalez**[1]                **Stuart Russell**[1]

[1]University of California, Berkeley
{tianjunz,paria.rashidinejad,jiantao,russell}@berkeley.edu

[2]Facebook AI Research
yuandong@fb.com

## Abstract

In online reinforcement learning (RL), efficient exploration remains particularly challenging in high-dimensional environments with sparse rewards. In low-dimensional environments, where tabular parameterization is possible, count-based upper confidence bound (UCB) exploration methods achieve minimax near-optimal rates. However, it remains unclear how to efficiently implement UCB in realistic RL tasks that involve nonlinear function approximation. To address this, we propose a new exploration approach via *maximizing* the deviation of the occupancy of the next policy from the explored regions. We add this term as an adaptive regularizer to the standard RL objective to trade off between exploration and exploitation. We pair the new objective with a provably convergent algorithm, giving rise to a new intrinsic reward that adjusts existing bonuses. The proposed intrinsic reward is easy to implement and combine with other existing RL algorithms to conduct exploration. As a proof of concept, we evaluate the new intrinsic reward on tabular examples across a variety of model-based and model-free algorithms, showing improvements over count-only exploration strategies. When tested on navigation and locomotion tasks from MiniGrid and DeepMind Control Suite benchmarks, our approach significantly improves sample efficiency over state-of-the-art methods.[2]

## 1 Introduction

Online RL is a useful tool for an agent to learn how to perform tasks, particularly when expert demonstrations are unavailable and reward information needs to be used instead [92]. To learn a satisfactory policy, an RL agent needs to effectively balance between exploration and exploitation, which remains a central question in RL [23, 15]. Exploration is particularly challenging in environments with sparse rewards. One popular approach to exploration is based on *intrinsic motivation*, often applied by adding an intrinsic reward (or bonus) to the extrinsic reward provided by the environment. In provable exploration methods, bonus often captures the value estimate uncertainty and the agent takes an action that maximizes the upper confidence bound (UCB) [5, 8, 41, 48, 44]. In tabular setting, UCB bonuses are often constructed based on either Hoeffding's inequality, which only uses visitation counts, or Bernstein's inequality, which uses value function variance in addition to visitation counts.

---

[*]Equal Contribution.
[2]Our code is available at https://github.com/tianjunz/MADE.

35th Conference on Neural Information Processing Systems (NeurIPS 2021).

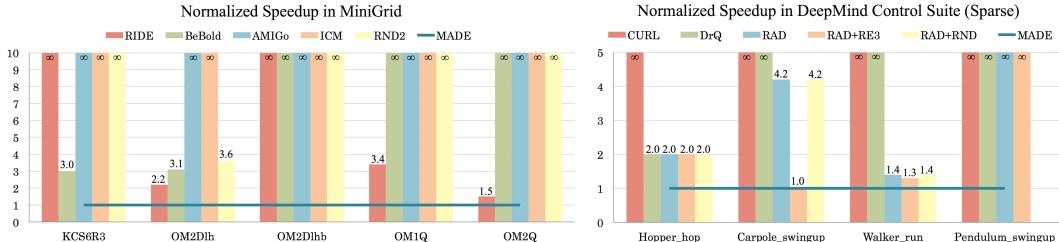

Figure 1: Normalized samples use of different methods with respect to MADE (smaller values are better). MADE consistency achieves a better sample efficiency compared to all other baselines. Infinity means the method fails to achieve maximum reward in given steps.

The latter is proved to be minimax near-optimal in environments with bounded rewards [44, 65] as well as bounded total reward [112] and reward-free settings [64, 49, 45, 113]. It remains an open question how one can efficiently compute confidence bounds to construct UCB bonus in non-linear function approximation. Furthermore, Bernstein-style bonuses are often hard to compute in practice beyond tabular setting, due to difficulties in computing value function variance.

In practice, various approaches are proposed to design intrinsic rewards: visitation pseudo-count bonuses estimate count-based UCB bonuses using function approximation [10, 15], curiosity-based bonuses seek states where model prediction error is high, uncertainty-based bonuses [77, 87] adopt ensembles of networks for estimating variance of the Q-function, empowerment-based approaches [52, 34, 84, 68] lead the agent to states over which the agent has control, and information gain bonuses [51] reward the agent based on the information gain between state-action pairs and next states.

Although the performance of practical intrinsic rewards is good in certain domains, empirically they are observed to suffer from issues such as detachment, derailment, and catastrophic forgetting [3, 23]. Moreover, these methods usually lack a clear objective and can get stuck in local optimum [3]. Indeed, the impressive performance currently achieved by some deep RL algorithms often revolves around manually designing dense rewards [13], complicated exploration strategies utilizing a significant amount of domain knowledge [23], or operating in the known environment regime [88, 69].

Motivated by current practical challenges and the gap between theory and practice, we propose a new algorithm for exploration by maximizing deviation from explored regions. This yields a practical algorithm with strong empirical performance. To be specific, we make the following contributions:

**1. Exploration via maximizing deviation** Our approach is based on modifying the standard RL objective (i.e. the cumulative reward) by adding a regularizer that adaptively changes across iterations. The regularizer can be a general function depending on the state-action visitation density and previous state-action coverage. We show that the regularized objective naturally admits several common existing exploration methods. We then choose a particular regularizer that **MA**ximizes the **DE**viation (MADE) of the next policy visitation $d^\pi$ from the regions covered by prior policies $\rho_{cov}^k$:

$$L_k(d^\pi) = J(d^\pi) + \tau_k \sum_{s,a} \sqrt{\frac{d^\pi(s,a)}{\rho_{cov}^k(s,a)}}. \tag{1}$$

Here, $k$ is the iteration number, $J(d^\pi)$ is the standard RL objective, and the regularizer encourages $d^\pi(s,a)$ to be large when $\rho_{cov}^k(s,a)$ is small. We give an algorithm for solving the regularized objective and prove that with access to an approximate planning oracle, it converges to the global optimum. We show that objective (1) results in an intrinsic reward that can be easily added to any RL algorithm to improve performance, as suggested by our empirical studies. Furthermore, the intrinsic reward applies a simple modification to the UCB-style bonus that considers prior visitation counts. This simple modification can also be added to existing bonuses in practice.

**2. Tabular studies** In the special case of tabular parameterization, we show that MADE only applies some simple adjustments to the Hoeffding-style count-based bonus. We compare the performance of MADE to Hoeffding and Bernstein bonuses in three different RL algorithms, for the exploration task in the stochastic diabolical bidirectional lock [3, 66], which has sparse rewards and local optima. Our results show that MADE robustly improves over the Hoeffding bonus and is competitive to the Bernstein bonus, across all three RL algorithms. Interestingly, MADE bonus and exploration strategy

appear to be very close to the Bernstein bonus, *without computing or estimating variance*, suggesting that MADE potentially captures some environmental structures. Additionally, we empirically show that MADE regularizer can improve the optimization rate in policy gradient methods.

**3. Experiments on MiniGrid and DeepMind Control Suite** We empirically show that MADE works well when combined with model-free (IMAPLA [25], RAD [55]) and model-based (Dreamer [35]) RL algorithms, greatly improving the sample efficiency over existing baselines. When tested in the procedurally-generated MiniGrid environments, MADE manages to converge with two to five times fewer samples compared to state-of-the-art method BeBold [111]. In DeepMind Control Suite [95], we build upon the model-free method RAD [55] and the model-based method Dreamer [35], improving the return up to 150 in 500K steps compared to baselines. Figure 1 shows normalized sample size to achieve maximum reward with respect to our algorithm.

## 2 Background

**Markov decision processes.** An infinite-horizon discounted MDP is described by a tuple $M = (\mathcal{S}, \mathcal{A}, P, r, \rho, \gamma)$, where $\mathcal{S}$ is the state space, $\mathcal{A}$ is the action space, $P : \mathcal{S} \times \mathcal{A} \mapsto \Delta(\mathcal{S})$ is the transition kernel, $r : \mathcal{S} \times \mathcal{A} \mapsto [0, 1]$ is the (extrinsic) reward function, $\rho : \mathcal{S} \mapsto \Delta(\mathcal{S})$ is the initial distribution, and $\gamma \in [0, 1)$ is the discount factor. A stationary (stochastic) policy $\pi \in \Delta(\mathcal{A} \mid \mathcal{S})$ specifies a distribution over actions in each state. Each policy $\pi$ induces a visitation density over state-action pairs $d^\pi : \mathcal{S} \times \mathcal{A} \mapsto [0, 1]$ defined as $d_\rho^\pi(s, a) := (1 - \gamma) \sum_{t=0}^\infty \gamma^t \mathbb{P}_t(s_t = s, a_t = a; \pi)$, where $\mathbb{P}_t(s_t = s, a_t = a; \pi)$ denotes $(s, a)$ visitation probability at step $t$, starting at $s_0 \sim \rho(\cdot)$ and following $\pi$. An important quantity is the value a policy $\pi$, which is the discounted sum of rewards $V^\pi(s) := \mathbb{E}[\sum_{t=0}^\infty \gamma^t r_t \mid s_0 = s, a_t \sim \pi(\cdot \mid s_t)$ for all $t \geq 0]$ starting at state $s \in \mathcal{S}$.

**Policy mixture.** For a sequence of policies $\mathcal{C}^k = (\pi_1, \ldots, \pi_k)$ with corresponding mixture distribution $w^k \in \Delta_{k-1}$, the policy mixture $\pi_{\text{mix},k} = (\mathcal{C}^k, w^k)$ is obtained by first sampling a policy from $w^k$ and then following that policy over subsequent steps [36]. The mixture policy induces a state-action visitation density according to $d^{\pi_{\text{mix}}}(s, a) = \sum_{i=1}^k w_i^k d^{\pi_i}(s, a)$. While the $\pi_{\text{mix}}$ may not be stationary in general, there exists a stationary policy $\pi'$ such that $d^{\pi'} = d^{\pi_{\text{mix}}}$ Puterman [80].

**Online reinforcement learning.** Online RL is the problem of finding a policy with a maximum value from an unknown MDP, using samples collected during exploration. Oftentimes, the following objective is considered, which is a scalar summary of the performance of policy $\pi$:

$$J_M(\pi) := \mathbb{E}_{s \sim \rho}[V^\pi(s)] = (1 - \gamma)^{-1} \mathbb{E}_{(s,a) \sim d_\rho^\pi(\cdot, \cdot)}[r(s, a)] = J(d^\pi). \tag{2}$$

We drop index $M$ when it is clear from context. We denote an optimal policy by $\pi^\star \in \arg\max_\pi J(\pi)$ and use the shorthand $V^\star := V^{\pi^\star}$ to denote the optimal value function. It is straightforward to check that $J(\pi)$ can equivalently be represented by the expectation of the reward over the visitation measure of $\pi$. We slightly abuse the notation and sometimes write $J(d^\pi)$ to denote the RL objective.

## 3 Adaptive regularization of the RL objective

### 3.1 Regularization to guide exploration

In online RL, the agent faces a dilemma in each step: whether it should select a seemingly optimal policy (exploit) or it should explore different regions of the MDP. To allow flexibility in this choice and trade off between exploration and exploitation, we propose to add a regularizer to the standard RL objective that changes throughout iterations of an online RL algorithm:

$$L_k(d^\pi) = \underbrace{J(d^\pi)}_{\text{exploitation}} + \tau_k \underbrace{R(d^\pi; \{d^{\pi_i}\}_{i=1}^k)}_{\text{exploration}}. \tag{3}$$

Here, $R(d^\pi; \{d^{\pi_i}\}_{i=1}^k)$ is a function of state-action visitation of $\pi$ as well as the visitation of prior policies $\pi_1, \ldots, \pi_k$. The temperature parameter $\tau_k$ determines the strength of regularization. Objective (3) is a *population* objective in the sense that it does not involve empirical estimations affected by the randomness in sample collection. In the following section, we give our particular choice of regularizer and discuss how this objective can describe some popular exploration bonuses. We then provide a convergence guarantee for the regularized objective in Section 3.2.

## 3.2 Exploration via maximizing deviation from policy cover

We develop our exploration strategy MADE based on a simple intuition: maximizing the deviation from the explored regions, i.e. all states and actions visited by prior policies. We define *policy cover* at iteration $k$ to be the density over regions explored by policies $\pi_1, \ldots, \pi_k$, i.e. $\rho^k_{\text{cov}}(s,a) := \frac{1}{k} \sum_{i=1}^{k} d^{\pi_i}(s,a)$. We then design our regularizer to encourage $d^\pi$ to be different from $\rho^k_{\text{cov}}$:

$$R_k(d^\pi; \{d^{\pi_i}\}_{i=1}^k) = \sum_{s,a} \sqrt{\frac{d^\pi(s,a)}{\rho^k_{\text{cov}}(s,a)}}. \tag{4}$$

It is easy to check that the maximizer of above function is $d^\pi(s,a) \propto \frac{1}{\rho^k_{\text{cov}}(s,a)}$. Our motivation behind this particular deviation is that it results in a simple modification of UCB bonus in tabular case.

We now compute the reward yielded by the new objective. First, define a policy mixture $\pi_{\text{mix},k}$ with policy sequence $(\pi_1, \ldots, \pi_k)$ and weights $((1-\eta)^{k-1}, (1-\eta)^{k-2}\eta, (1-\eta)^{k-3}\eta, \ldots, \eta)$ for $\eta > 0$. Let $d^{\pi_{\text{mix},k}}$ be the visitation density of $\pi_{\text{mix},k}$. We compute the total reward at iteration $k$ by taking the gradient of the new objective with respect to $d^\pi$ at $d^{\pi_{\text{mix},k}}$:

$$r_k(s,a) = (1-\gamma)\nabla_d L_k(d)\big|_{d=d^{\pi_{\text{mix},k}}} = r(s,a) + (1-\gamma)\tau_k \nabla_d R_k(d; \{d^{\pi_i}\}_{i=1}^k)\big|_{d=d^{\pi_{\text{mix},k}}}, \tag{5}$$

which gives the following reward

$$r_k(s,a) = r(s,a) + \frac{(1-\gamma)\tau_k/2}{\sqrt{d^{\pi_{\text{mix},k}}(s,a)\rho^k_{\text{cov}}(s,a)}}. \tag{6}$$

The intrinsic reward is constructed based on two densities: $\rho^k_{\text{cov}}$ a uniform combination of past visitation densities and $\hat{d}^{\pi_{\text{mix},k}}$ a weighted mixture of the past visitation densities. As we will discuss shortly, policy cover $\rho^k_{\text{cov}}(s,a)$ is related to the visitation count of $(s,a)$ pair in previous iterations and resembles count-based bonuses [10, 44] or their approximates such as RND [15]. Therefore, for an appropriate choice of $\tau_k$, MADE intrinsic reward decreases as the number of visitations increases.

MADE intrinsic reward is also proportional to $1/\sqrt{d^{\pi_{\text{mix},k}}(s,a)}$, which can be viewed as a correction applied to the count-based bonus. In effect, due to the decay of weights in $\pi_{\text{mix},k}$, the above construction gives a higher reward to $(s,a)$ pairs visited earlier. Experimental results suggest that this correction may alleviate major difficulties in sparse reward exploration, namely detachment and catastrophic forgetting, by encouraging the agent to revisit forgotten states and actions.

Empirically, MADE's intrinsic reward is computed based on estimates $\hat{d}^{\pi_{\text{mix},k}}$ and $\hat{\rho}^k_{\text{cov}}$ from data collected by iteration $k$. Furthermore, practically we consider a smoothed version of the above regularizer by adding $\lambda > 0$ to both numerator and denominator; see Equation (7).

**MADE intrinsic reward in tabular case.** In tabular empirical setting, the empirical estimation of policy cover is simply $\hat{\rho}^k_{\text{cov}}(s,a) = \frac{N_k(s,a)}{N_k}$, where $N_k(s,a)$ is $(s,a)$ pair's visitation count and $N_k$ is the total count, by iteration $k$. Setting $\tau_k = 1/\sqrt{N_k}$, MADE simply modifies the Hoeffding-type bonus via the mixture density and is proportional to $1/\sqrt{\hat{d}^{\pi_{\text{mix},k}}(s,a)N_k(s,a)}$.

Bernstein bonus is another tabular UCB bonus that modifies Hoeffding bonus via an empirical estimate of the value function variance. Bernstein bonus is shown to improve over Hoeffding count-only bonus by exploiting additional environment structure [106] and close the gap between algorithmic upper bounds and information-theoretic limits up to logarithmic factors [112, 113]. However, a practical and efficient implementation of a bonus that exploits variance information in non-linear function approximation parameterization still remains an open question; see Section 6 for further discussion. On the other hand, our proposed modification based on the mixture density can be easily and efficiently incorporated with non-linear parameterization.

**Deriving some popular bonuses from regularization.** The regularization in (3) can describe some popular bonuses. Exploration bonuses that only depend on state-action visitation counts can be expressed in the form (3) by setting the regularizer a linear function of $d^\pi$ and the exploration bonus $r_i(s,a)$, i.e., $R_k(d^\pi; \{d^{\pi_i}\}_{i=1}^k) = \sum_{s,a} d^\pi(s,a)r_i(s,a)$. One can check that taking the gradient of the regularizer with respect to $d^\pi$ recovers $r_i(s,a)$. As another example, one can set the regularizer to Shannon entropy $R_k(d^\pi; \{d^{\pi_i}\}_{i=1}^k) = -\sum_{s,a} d^\pi(s,a) \log d^\pi(s,a)$, which gives the intrinsic reward $-\log d^\pi(s,a)$ (up to an additive constant) and recovers the result in Zhang et al. [109].

---

**Algorithm 1** Policy computation for adaptively regularized objective

---

1: **Inputs:** Iteration count $K$, planning error $\epsilon_p$, visitation density error $\epsilon_d$.
2: Initialize policy mixture $\pi_{\text{mix},1} =$ with $\mathcal{C}_1 = (\pi_1)$ and $w^1 = (1)$
3: **for** $k = 1, \ldots, K$ **do**
4:      Estimate the visitation density $\hat{d}^{\pi_{\text{mix},k}}$ of $\pi_{\text{mix},k}$ via a visitation density oracle.
5:      Compute reward $r_k(s,a) = r(s,a) + (1-\gamma)\tau_k \nabla_d R_k(d; \{\pi_i\}_{i=1}^k)\big|_{d=\hat{d}^{\pi_{\text{mix},k}}}$.
6:      Run approximate planning on modified MDP $M^k = (\mathcal{S}, \mathcal{A}, P, r_k, \gamma)$ and return $\pi_{k+1}$.
7:      Update policy mixture $\mathcal{C}^{k+1} = (C_k, \pi_{k+1})$ and $w^{k+1} = ((1-\eta)w^k, \eta)$.
8: **Return:** $\pi_{\text{mix},K} = (\mathcal{C}^k, w^k)$.

---

### 3.3 Solving the regularized objective

Recall that our goal is to find a policy that maximizes the regularized objective $\pi^\star \in \arg\max_\pi L_k(d^\pi) = J(d^\pi) + \tau_k R(d^\pi; \{d^{\pi_i}\}_{i=1}^k)$. Despite being nonconcave in $\pi$, this objective is concave in $d^\pi$. Therefore, one can solve the following constrained concave optimization problem instead: $\max_{d^\pi \in \mathcal{V}} L_k(d^\pi)$, where $\mathcal{V}$ is the set of all valid visitation densities.

To solve this constrained optimization problem, we use the conditional gradient method or the Frank-Wolfe algorithm [31], which has been used in the context of RL in works such as [36, 1, 105]. The conditional gradient method involves iteratively solving $d^{\pi_{k+1}} \in \arg\max_d \langle d, \nabla_{d^\pi} L_k(d^\pi)\big|_{d^\pi = d^{\pi_{\text{mix},k}}} \rangle$ and obtaining $d^{\pi_{\text{mix},k+1}}$ from a weighted combination of $d^{\pi_{\text{mix},k}}$ and $d^{\pi_{k+1}}$. Note that the first step is equivalent to planning via a reward function proportional to $\nabla_{d^\pi} L_k(d^\pi)\big|_{d^\pi = d^{\pi_{\text{mix},k}}}$, which justifies they way we computed the total reward in Equation (5).

We provide convergence guarantee for Algorithm 1 in the following theorem whose proof is given in Appendix A.

**Theorem 1.** *Consider the following regularizer for (3) with $\lambda > 0$ and a valid visitation density $d$*

$$R_\lambda(d; \{d^{\pi_i}\}_{i=1}^k) = \sum_{s,a} \sqrt{\frac{d(s,a) + \lambda}{\rho_{cov}^k(s,a) + \lambda}}, \tag{7}$$

*Set $\tau_k = \tau/k^c$, where $0 < \tau < 1$ and $c > 0$. For any $\epsilon > 0$ and $\eta \leq 0.4\epsilon\lambda^2$, there exists $\epsilon_p, \epsilon_d, c, B$ such that $\pi_{\text{mix},K}$ returned by Algorithm 1 after $K \geq \eta^{-1}\log(10B\epsilon^{-1})$ iterations satisfies $L_k(d^{\pi_{\text{mix},K}}) \geq \max_\pi L_k(d^\pi) - \epsilon$.*

**Remark 1.** *One does not need to maintain the functional forms of past policies to estimate $\hat{d}^{\pi_{\text{mix},k}}$. Practically, one may truncate the dataset to a (prioritized) buffer and estimate the density over that.*

## 4 A tabular study

We first study the performance of MADE in tabular toy examples. In the Bidirectional Lock experiment, we compare MADE to theoretically guaranteed Hoeffding-style and Bernstein-style bonuses in a sparse reward exploration task. In the Chain MDP, we investigate whether MADE's regularizer (4) provides any benefits in improving optimization rate in policy gradient methods.

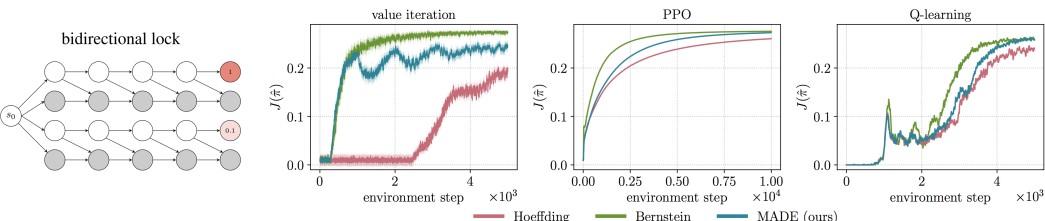

Figure 2: In a bidirectional lock, the agent starts at $s_0$ and enters one of the chains based on the selected action. Each chain has a positive reward at the end, $H$ good states, and $H$ dead states. Both actions available to the agent lead it to the dead state, one with probability one and the other with probability $p < 1$. MADE performs better than Hoeffding-style bonus and comparable to Bernstein-style bonus across all three RL algorithms.

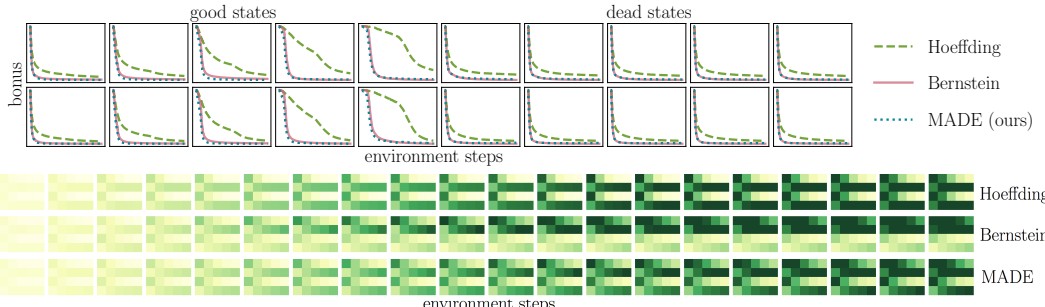

Figure 3: Hoeffding, Bernstein, and MADE intrinsic rewards over iterations in a bidirectional lock.

## 4.1 Exploration in bidirectional lock

We consider a stochastic version of the bidirectional combination lock (Figure 2), which is considered a particularly difficult exploration task in tabular setting [66, 3]. This environment is challenging because: (1) positive rewards are sparse, (2) a small negative reward is given when transiting to a good state and thus, moving to a dead state is locally optimal, and (3) the agent may forget to explore one chain and get stuck in local minima upon receiving an end reward in one lock [3].

**RL algorithms and exploration strategies.** We compare the performance of Hoeffding and Bernstein bonuses [44] to MADE in three different RL algorithms. To implement MADE in tabular setting, we simply use two buffers: one that stores all past state-action pairs to estimate $\rho_{cov}$ and another one that only maintains the most recent $B$ pairs to estimate $d_\mu^\pi$. We use empirical counts to estimate both densities, which give a bonus $\propto 1/\sqrt{N_k(s,a)B_k(s,a)}$, where $N_k(s,a)$ is the total count and $B_k(s,a)$ is the recent buffer count of $(s,a)$ pair. We combine three bonuses with three RL algorithms: (1) value iteration with bonus [37], (2) proximal policy optimization (PPO) with a model [16], and (3) Q-learning with bonus [44].

**Results.** Figure 2 summarizes our results showing MADE improves over the Hoeffding bonus and is competitive to the Bernstein bonus in all three algorithms. Unlike Bernstein bonus that is hard to compute beyond tabular setting, MADE bonus design is simple and can be effectively combined with any deep RL algorithm. The experimental results suggest several interesting properties for MADE. First, MADE applies a simple modification to the Hoeffding bonus which improves the performance. Second, as illustrated in Figure 3, bonus values and exploration pattern of MADE is somewhat similar to the Bernstein bonus. This suggests that MADE may capture some structural information of the environment, similar to Bernstein bonus, which captures certain environmental properties such as the degree of stochasticity [106].

## 4.2 Policy gradient in chain MDP

We consider the chain MDP (Figure 4) presented in [2], which suffers from vanishing gradients with policy gradient approach [93] as a positive reward is only achieved if the agent always takes action $a_1$. This leads to an exponential iteration complexity lower bound on the convergence of vanilla policy

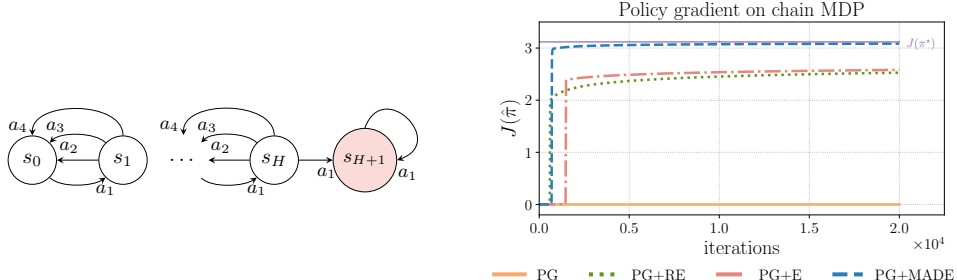

Figure 4: A deterministic chain MDP that suffers from vanishing gradients [2]. We consider a constrained tabular policy parameterization with $\pi(a|s) = \theta_{s,a}$ and $\sum_a \theta_{s,a} = 1$. The agent always starts from $s_0$ and the only non-zero reward is $r(s_{H+1}, a_1) = 1$.

gradient approach even with access to exact gradients [2]. In this environment the agent always starts at state $s_0$ and recent guarantees on the global convergence of exact policy gradients are vacuous [11, 2, 62]. This is because the rates depend on the ratio between the optimal and learned visitation densities, known as *concentrability coefficient* [47, 85, 32, 82], or the ratio between the optimal visitation density and initial distribution [2].

**RL algorithms.**    Since our goal in this experiment is to investigate the optimization effects and not exploration, we assume access to exact gradients. In this setting, we consider MADE regularizer with the form $\sum_{s,a} \sqrt{d^\pi(s,a)}$. Note that policy gradients take gradient of the objective with respect to the policy parameters $\theta$ and not $d^\pi$. We compare optimizing the policy gradient objective with four methods: vanilla version PG (e.g. uses policy gradient theorem [99, 93, 53]), relative policy entropy regularization PG+RE [2], policy entropy regularization PG+E [67, 62], and MADE regularization.

**Results.**    Figure 4 illustrates our results on policy gradient methods. As expected [2], the vanilla version has a very slow convergence rate. Both entropy and relative entropy regularization methods are proved to achieve a linear convergence rate of $\exp(-t)$ in the iteration count $t$ [62, 2]. Interestingly, MADE seems to outperforms the policy entropy regularizers, quickly converging to a globally optimal policy.

## 5    Experiments on MiniGrid and DeepMind Control Suite

In addition to the tabular setting, MADE can also be integrated with various model-free and model-based deep RL algorithms such as IMPALA [25], RAD [57], and Dreamer [35]. As we will see shortly, MADE exploration strategy on MiniGrid [19] and DeepMind Control Suite [95] tasks achieves state-of-the-art sample efficiency.

For a practical estimation of $\rho_{\text{cov}}^k$ and $d^{\pi_{\text{mix},k}}$, we adopt the two buffer idea described in the tabular setting. However, since now the state space is high-dimensional, we use RND [15] to estimate $N_k(s,a)$ (and thus $\rho_{\text{cov}}^k$) and use a variational auto-encoder (VAE) to estimate $d^{\pi_{\text{mix},k}}$. Specifically, for RND, we minimize the difference between a predictor network $\phi'(s,a)$ and a randomly initialized target network $\phi(s,a)$ and train it in an online manner as the agent collects data. We sample data from the recent buffer $\mathcal{B}$ to train a VAE. The length of $\mathcal{B}$ is a design choice for which we do an ablation study. Thus, the intrinsic reward in deep RL setting takes the following form

$$(1-\gamma)\tau_k \frac{\|\phi(s,a) - \phi'(s,a)\|}{\sqrt{d^{\pi_{\text{mix},k}}(s,a)}}.$$

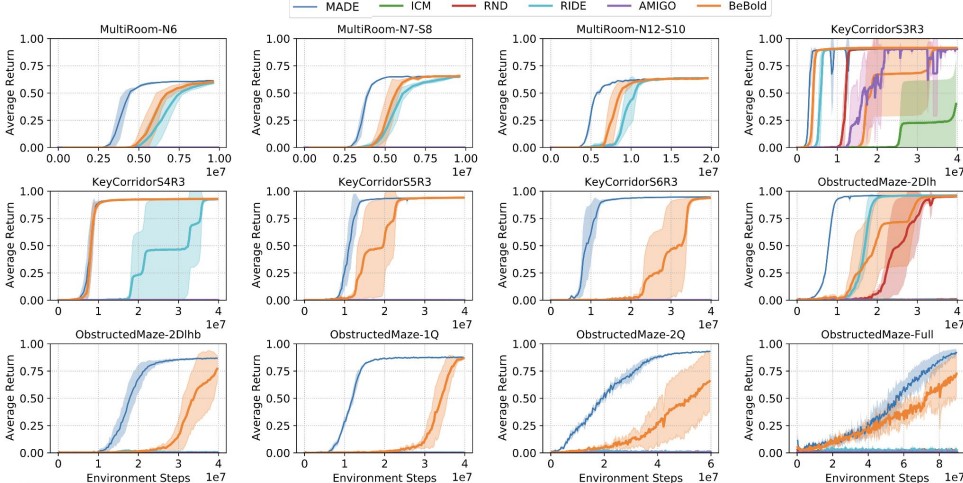

Figure 5: Results for various hard exploration tasks from MiniGrid. MADE successfully solves all the environments while other algorithms (except for BeBold) fail to solve several environments. MADE finds the optimal solution with 2-5 times fewer samples, yielding a much better sample efficiency.

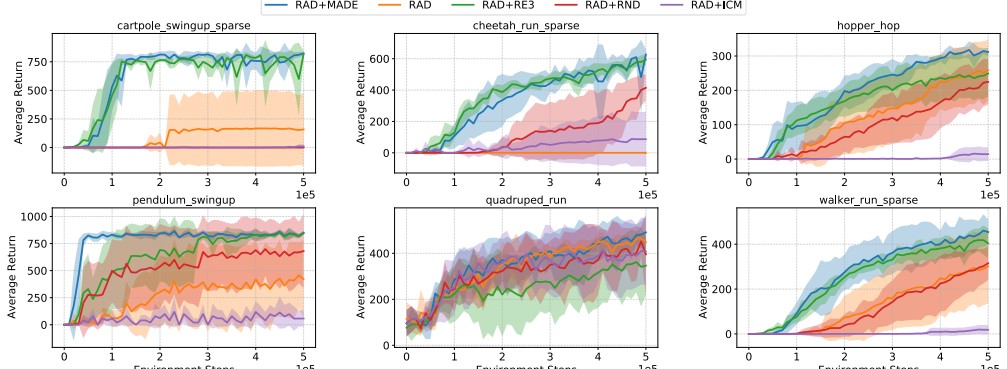

Figure 6: Results for several DeepMind control suite locomotion tasks. Comparing to all baselines, the performance of MADE is consistently better. Sometimes baseline methods even fail to solve the task.

**Model-free RL baselines.** We consider several baselines in MiniGrid: **IMPALA** [25] is a variant of policy gradient algorithms which we use as the training baseline; **ICM** [76] learns a forward and reverse model for predicting state transition and uses the forward model prediction error as intrinsic reward; **RND** [15] trains a predictor network to mimic a randomly initialized target network as discussed above; **RIDE** [81] learns a representation similar to ICM and uses the difference of learned representations along a trajectory as intrinsic reward; **AMIGo** [17] learns a teacher agent to assign intrinsic reward; **BeBold** [111] adopts a regulated difference of novelty measure using RND. In DeepMind Control Suite, we consider **RE3** [86] as a baseline which uses a random encoder for state embedding followed by a $k$-nearest neighbour bonus for a maximum state coverage objective.

**Model-based RL baselines.** MADE can be combined with model-based RL algorithms to improve sample efficiency. For baselines, we consider **Dreamer**, which is a well-known model-based RL algorithm for DeepMind Control Suite, as well as **Dreamer+RE3**, which includes RE3 bonus on top of Dreamer.

MADE achieves state-of-the-art results on both navigation and locomotion tasks by a substantial margin, greatly improving the sample efficiency of the RL exploration in both model-free and model-based methods. Details on experiments and hyperparameters are provided in Appendix B.

## 5.1 Model-free RL on MiniGrid

MiniGrid [19] is a widely used benchmark for exploration in RL. Despite having symbolic states and a discrete action space, MiniGrid tasks are quite challenging. The easiest task is **MultiRoom** (MR) in which the agent needs to navigate to the goal by going to different rooms connected by the doors. In **KeyCorridor** (KC), the agent needs to search around different rooms to find the key and then use it to open the door. **ObstructedMaze** (OM) is a harder version of KC where the key is hidden in a box and sometimes the door is blocked by an obstruct. In addition to that, the entire environment is procedurally-generated. This adds another layer of difficulty to the problem.

From Figure 5 we can see that MADE manages to solve all the challenging tasks within 90M steps while all other baselines (except BeBold) only solve up to 50% of them. Compared to BeBold, MADE uses significantly (2-5 times) fewer samples.

## 5.2 Model-free RL on DeepMind Control

We also test MADE on image-based continuous control tasks of DeepMind Control Suite [95], which is a collection of diverse control tasks such as Pendulum, Hopper, and Acrobot with realistic simulations. Compared to MiniGrid, these tasks are more realistic and complex as they involve stochastic transitions, high-dimensional states, and continuous actions. For baselines, we build our algorithm on top of RAD [57], a strong model-free RL algorithm with a competitive sample efficiency. We compare our approach with ICM, RND, as well as RE3, which is the SOTA algorithm.[3] Note that we compare MADE to very strong baselines. Other algorithms such as DrQ [54], CURL [89], ProtoRL [104], SAC+AE [103]) perform worse based on the results reported in the original papers.

---

[3]As we were not provided with the source code, we implemented ICM and RND ourselves. The performance for ICM is slightly worse than what the author reported, but the performance of RND and RE3 is similar.

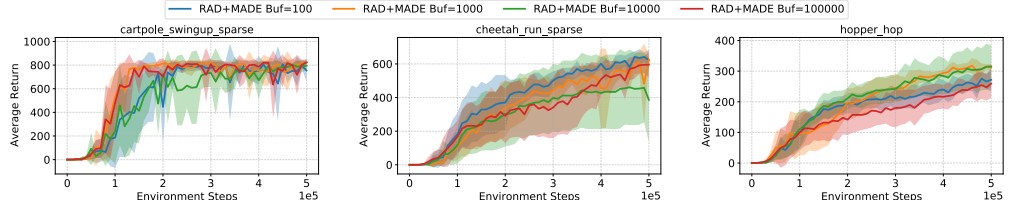

Figure 7: Ablation study on buffer size in MADE. The optimal buffer size varies in different tasks. We found that a buffer size of 10000 empirically works consistently reasonable.

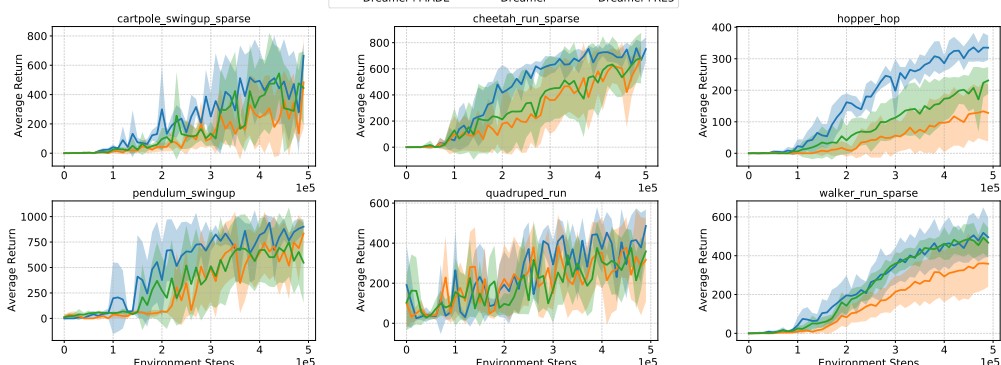

Figure 8: Results for DeepMind control suite locomotion tasks in model-based RL setting. Comparing to all baselines, the performance of MADE is consistently better. Some baseline methods even fail to solve the task.

MADE show consistent improvement in sample efficiency: 2.6 times over RAD+RE3, 3.3 times over RAD+RND, 19.7 times over CURL, 15.0 times over DrQ and 3.8 times over RAD.

From Figure 6, we can see that MADE consistently improves sample efficiency compared to all baselines. For these tasks, RND and ICM do not perform well and even fail on `Cartpole-Swingup`. RE3 achieves a comparable performance in two tasks, however, its performance on `Pendulum-Swingup`, `Quadruped-Run`, `Hopper-Hop` and `Walker-Run` is significantly worse than MADE. For example, in `Pendulum-Swingup`, MADE achieves a reward of around 800 in only 30K steps while RE3 requires 300k samples. In `Quadruped-Run`, there is a 150 reward gap between MADE and RE3, which seems to be still enlarging. These tasks show the strong performance of MADE in model-free RL.

**Ablation study.** We study how the buffer length affects the performance of our algorithm in some DeepMind Control tasks. Results show that for different tasks the optimal length is slightly different. We empirically found that using a buffer length of 1000 consistently works well across different tasks.

### 5.3 Model-based RL on DeepMind Control

We also empirically verify the performance of MADE combined with the SOTA model-based RL algorithm Dreamer [35]. We compare MADE with Dreamer and Dreamer combined with RE3 in Figure 8. Results show that MADE has great sample efficiency in maps like `Cheetah-Run-Sparse`, `Hopper-Hop` and `Pendulum-Swingup`. For example, in `Hopper-Hop`, MADE achieves more than 100 higher return than RE3 and 250 higher than Dreamer, achieving a new SOTA result.

## 6 Related work

**Provable optimistic exploration.** Most provable exploration strategies are based on optimism in the face of uncertainty (OFU) principle. In tabular setting, model-based exploration algorithms include variants of UCB [50, 12], UCRL [56, 41, 106, 49, 64], and Thompson sampling [101, 5, 83] and value-based methods include optimistic Q-learning [44, 98, 90, 60, 65] and value-iteration with UCB [8, 112, 113, 45]. These methods are recently extended to linear MDP setting leading to a variety of model-based [114, 7, 42, 115], value-based [97, 46], and policy-based algorithms [16, 108, 3]. Going beyond linear function approximation, systematic exploration strategies are developed based on structural assumptions on MDP such as low Bellman rank [43] and block MDP [22]. These methods are either computationally intractable [43, 91, 7, 107, 102, 21, 96] or are only oracle efficient

[26, 4]. The recent work [27] provides a sample efficient approach with non-linear policies, however, the algorithm requires maintaining the functional form of all prior policies.

**Practical exploration via intrinsic reward.** Apart from previously-discussed methods, other works give intrinsic reward based on the difference in (abstraction of) consecutive states [110, 61, 81]. However, this approach is inconsistent: the intrinsic reward does not converge to zero and thus, even with infinite samples, the final policy does not maximize the RL objective. Other intrinsic rewards try to estimate pseudo-counts [10, 94, 15, 14, 75, 9], inspired by provable count-based methods. Though favoring novel states, practically these methods might suffer from *detachment and derailment* [23, 24], and *forgetting* [3]. More recent works propose a combination of different criteria. RIDE [81] learns a representation using curiosity criterion and uses the difference of consecutive states along the trajectory as the bonus. AMIGo [17] learns a teacher agent for assigning rewards for exploration. Go-Explore [23] explicitly decouples the exploration and exploitation stage, yields a more sophisticated algorithm with many hand-tuned hyperparameters. Prior work also tries to add a bonus on maximizing the KL-divergence between the current policy and the previous policies [39]. However, we emphasize that this is indeed an entirely different principle compared with ours, simply because difference of visitation density of the policies is not equivalent to the KL-divergence between policies. In another word, two significantly different policies do not necessarily induce significantly different visitation densities and vice versa.

**Maximum entropy exploration.** Another line of work encourages exploration via maximizing some type of entropy. One category maximizes policy entropy [67] or relative entropy [2] in addition to the RL objective. The work [28] modifies the RL objective by introducing an adversarial policy which results in the next policy to move away from prior policies while staying close to the current policy. In contrast, our approach focuses on the regions explored by prior policies as opposed to the prior policies themselves. Recently, effects of policy entropy regularization have been studied theoretically [72, 33]. In policy gradient methods with access to exact gradients, policy entropy regularization results in faster convergence by improving the optimization landscape [62, 63, 6, 18]. Another category considers maximizing the entropy of state or state-action visitation densities such as Shannon entropy [36, 40, 58, 86] or Rényi entropy [109]. Empirically, our approach achieves better performance over entropy-based methods.

**Other exploration strategies.** Besides intrinsic motivation, other strategies are also fruitful in encouraging the RL agent to visit a wide range of states. One example is exploration by injecting noise to the action action space [59, 73, 38, 74] or parameter space [30, 78]. Another example is the reward-shaping category, in which diverse goals are set to guide exploration [20, 29, 71, 79].

# 7 Discussion

We introduce a new exploration strategy MADE based on maximizing deviation from explored regions. We show that by simply adding a regularizer to the original RL objective, we get an easy-to-implement intrinsic reward which can be incorporated with any RL algorithm. We provide a policy computation algorithm for this objective and prove that it converges to a global optimum, provided that we have access to an approximate planner. In tabular setting, MADE consistently improves over the Hoeffding bonus and shows competitive performance to the Bernstein bonus, while the latter is impractical to compute beyond tabular. We conduct extensive experiments on MiniGrid, showing a significant (over 5 times) reduction of the required sample size. MADE also performs well in DeepMind Control Suite when combined with both model-free and model-based RL algorithms, achieving SOTA sample efficiency results. One limitation of the current work is that it only uses the naive representations of states (e.g., one-hot representation in tabular case). In fact, exploration could be conducted much more efficiently if MADE is implemented with a more compact representation of states. We leave this direction to future work.

# 8 Acknowledgements

The authors are grateful to Andrea Zanette for helpful discussions. The authors thank Alekh Agarwal, Michael Henaff, Sham Kakade, and Wen Sun for providing their code. Paria Rashidinejad is partially supported by the Open Philanthropy Foundation, Intel, and the Leverhulme Trust. Jiantao Jiao is partially supported by NSF grants IIS-1901252, CCF-1909499, and DMS-2023505. Tianjun Zhang is supported by the BAIR Commons at UC-Berkeley and thanks Commons sponsors for their support. In

addition to NSF CISE Expeditions Award CCF-1730628, UC Berkeley research is supported by gifts from Alibaba, Amazon Web Services, Ant Financial, CapitalOne, Ericsson, Facebook, Futurewei, Google, Intel, Microsoft, Nvidia, Scotiabank, Splunk and VMware.

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
