# A Convergence analysis of Algorithm 1

In this section, we provide a convergence rate analysis for Algorithm 1. Similar to Hazan et al. [36], Algorithm 1 has access to an approximate density oracle and an approximate planner defined below:

- *Visitation density oracle:* We assume access to an approximate density estimator that takes in a policy $\pi$ and a density approximation error $\epsilon_d \geq 0$ as inputs and returns $\hat{d}^\pi$ such that $\|d^\pi - \hat{d}^\pi\|_\infty \leq \epsilon_d$.
- *Approximate planning oracle:* We assume access to an approximate planner that, given any MDP $M$ and error tolerance $\epsilon_p \geq 0$, returns a policy $\pi$ such that $J_M(\pi) \geq \max_\pi J_M(\pi) - \epsilon_p$.

## A.1 Proof of Theorem 1

We first give the following proposition that captures certain properties of the proposed objective. The proof is postponed to the end of this section.

**Proposition 1.** *Consider the following regularization for $\lambda > 0$*

$$R_\lambda(d; \{d^{\pi_i}\}_{i=1}^k) = \sum_{s,a} \sqrt{\frac{d(s,a) + \lambda}{\rho_{cov}(s,a) + \lambda}},$$

*with $\tau_k = \tau/k^c$ where $\tau < 1, c > 0$. There exist constants $\beta, B,$ and $\xi$ that only depend on MDP parameters and $\lambda$ such that $L_k(d) := J(d) + \tau_k R_\lambda(d; \{d^{\pi_i}\}_{i=1}^k)$ satisfies the following regularity conditions for all $k \geq 1$, an appropriate choice of $c$, and valid visitation densities $d$ and $d'$:*

*(i) $L_k(d)$ is concave in $d$;*

*(ii) $L_k(d)$ is $\beta$-smooth: $\|\nabla L_k(d) - \nabla L_k(d')\|_\infty \leq \beta \|d - d'\|_\infty, \quad -\beta I \preceq \nabla^2 L_k(d) \preceq \beta I$;*

*(iii) $L_k(d)$ is $B$-bounded: $L_k(d) \leq B, \quad \|\nabla L_k(d)\|_\infty \leq B$;*

*(iv) There exists $\delta_k$ such that $\max_d L_{k+1}(d) - L_k(d) \leq \delta_k$ and we have $\sum_{i=0}^k (1-\eta)^i \delta_{k-i} \leq \tau\xi$.*

Taking the above proposition as given for the moment, we prove Theorem 1 following steps similar to those of Hazan et al. [36, Theorem 4.1]. By construction of the mixture density $d^{\pi_{\text{mix},k}}$, we have

$$d^{\pi_{\text{mix},k}} = (1-\eta)d^{\pi_{\text{mix},k-1}} + \eta d^{\pi_k}.$$

Combining the above equation with the $\beta$-smoothness of $L_k(d)$ yields

$$L_k(d^{\pi_{\text{mix},k}}) = L_k((1-\eta)d^{\pi_{\text{mix},k-1}} + \eta d^{\pi_k})$$
$$\geq L_k(d^{\pi_{\text{mix},k-1}}) + \eta\langle d^{\pi_k} - d^{\pi_{\text{mix},k-1}}, \nabla L_k(d^{\pi_{\text{mix},k-1}})\rangle - \eta^2\beta\|d^{\pi_k} - d^{\pi_{\text{mix},k-1}}\|_2^2$$
$$\geq L_k(d^{\pi_{\text{mix},k-1}}) + \eta\langle d^{\pi_k} - d^{\pi_{\text{mix},k-1}}, \nabla L_k(d^{\pi_{\text{mix},k-1}})\rangle - 4\eta^2\beta. \tag{8}$$

Here the last inequality uses $\|d^{\pi_k} - d^{\pi_{\text{mix},k-1}}\|_2 \leq 2$. By property (ii), we bound $\langle d^{\pi_k}, \nabla L_k(d^{\pi_{\text{mix},k-1}})\rangle$ according to

$$\langle d^{\pi_k}, \nabla L_k(d^{\pi_{\text{mix},k-1}})\rangle \geq \langle d^{\pi_k}, \nabla L_k(\hat{d}^{\pi_{\text{mix},k-1}})\rangle - \beta\|d^{\pi_{\text{mix},k-1}} - \hat{d}^{\pi_{\text{mix},k-1}}\|_\infty$$
$$\geq \langle d^{\pi_k}, \nabla L_k(\hat{d}^{\pi_{\text{mix},k-1}})\rangle - \beta\epsilon_d, \tag{9}$$

where in the last step we used the density oracle approximation error. Recall that we defined $r_k = (1-\gamma)\nabla L_k(\hat{d}^{\pi_{\text{mix},k-1}})$. Since $\pi_k$ returned by the approximate planning oracle is an $\epsilon_p$-optimal policy in $M^k$, we have $(1-\gamma)^{-1}\langle d^{\pi_k}, r_k\rangle \geq (1-\gamma)^{-1}\langle d^\pi, r_k\rangle - \epsilon_p$ for any policy $\pi$, including $\pi^\star$. Therefore,

$$\langle d^{\pi_k}, \nabla L_k(d^{\pi_{\text{mix},k-1}})\rangle \geq \langle d^{\pi^\star}, \nabla L_k(\hat{d}^{\pi_{\text{mix},k-1}})\rangle - \epsilon_p - \beta\epsilon_d$$
$$\geq \langle d^{\pi^\star}, \nabla L_k(d^{\pi_{\text{mix},k-1}})\rangle - \epsilon_p - 2\beta\epsilon_d, \tag{10}$$

where we used the density oracle approximation error once more in the second step. Going back to inequality (8), we further bound $L_k(d^{\pi_{\text{mix},k}})$ by

$$L_k(d^{\pi_{\text{mix},k}}) \geq L_k(d^{\pi_{\text{mix},k-1}}) + \eta\langle d^{\pi_k} - d^{\pi_{\text{mix},k-1}}, \nabla L_k(d^{\pi_{\text{mix},k-1}})\rangle - 4\eta^2\beta$$
$$\geq L_k(d^{\pi_{\text{mix},k-1}}) + \eta\langle d^{\pi^\star} - d^{\pi_{\text{mix},k-1}}, \nabla L_k(d^{\pi_{\text{mix},k-1}})\rangle - \eta\epsilon_p - 2\eta\beta\epsilon_d - 4\eta^2\beta$$
$$\geq (1-\eta)L_k(d^{\pi_{\text{mix},k-1}}) + \eta L_k(d^{\pi^\star}) - 4\eta^2\beta - \eta\epsilon_p - 2\eta\beta\epsilon_d,$$

where the last inequality is by concavity of $L_k(d)$. Therefore,

$$L_k(d^{\pi^\star}) - L_k(d^{\pi_{\mathrm{mix},k}}) \le (1-\eta)[L_k(d^{\pi^\star}) - L_k(d^{\pi_{\mathrm{mix},k-1}})] + 2\eta\beta\epsilon_d + \eta\epsilon_p + 4\eta^2\beta.$$

By assumption (iv), we write

$$
\begin{aligned}
L_{K+1}(d^{\pi^\star}) - L_{K+1}(d^{\pi_{\mathrm{mix},K}}) &\le L_K(d^{\pi^\star}) - L_K(d^{\pi_{\mathrm{mix},K}}) + 2\delta_K \\
&\le (1-\eta)[L_K(d^{\pi^\star}) - L_K(d^{\pi_{\mathrm{mix},K-1}})] + 2\delta_K + 2\eta\beta\epsilon_d + \eta\epsilon_p + 4\eta^2\beta \\
&\le Be^{-\eta K} + 2\beta\epsilon_d + \epsilon_p + 4\eta\beta + 2\sum_{i=0}^{K}(1-\eta)^i \delta_{K-i} \\
&\le Be^{-\eta K} + 2\beta\epsilon_d + \epsilon_p + 4\eta\beta + 2\tau\xi.
\end{aligned}
$$

It is straightforward to check that setting $\eta \le 0.1\epsilon\beta^{-1}, \epsilon_p \le 0.1\epsilon, \epsilon_d \le 0.1\epsilon\beta^{-1}, \tau \le 0.1\epsilon$, and the number of iterations $K \ge \eta^{-1}\log(10B\epsilon^{-1})$ yields the claim of Theorem 1.

**Remark 2.** *Since the temperature parameter $\tau_k$ in Proposition 1 goes to zero as $k$ increases, one can show that the expected value of policy returned by Algorithm 1 converges to the maximum performance $J(\pi^\star)$.*

*Proof of Proposition 1.* For claim (ii), observe that $\nabla^2 L_k(d)$ is a diagonal matrix whose $(s,a)$ diagonal term is given by

$$(\nabla^2 L_k(d))_{s,a} = \frac{-\tau}{4k^c} \times \frac{1}{(d(s,a)+\lambda)^{3/2}(\rho_{\mathrm{cov}}(s,a)+\lambda)^{1/2}}.$$

The diagonal elements are bounded by $-1/(4\lambda^2) \le (\nabla^2 L_k(d))_{s,a} \le \frac{1}{4\lambda^2} =: \beta$. Furthermore, by Taylor's theorem, one has

$$\|\nabla L_k(d) - \nabla L_k(d')\|_\infty \le \max_{(s,a),\alpha\in[0,1]}(\nabla^2 L_k(\alpha d + (1-\alpha)d'))\|d-d'\|_\infty \le \beta\|d-d'\|_\infty.$$

Claim (i) is immediate from the above calculation as the Hessian $\nabla^2 L_k(d)$ is negative definite. Claim (iii) may be verified by explicit calculation:

$$\sum_{s,a} d(s,a)r(s,a) + \frac{\tau}{k^c}\sum_{s,a}\sqrt{\frac{d(s,a)+\lambda}{\rho_{\mathrm{cov}}(s,a)+\lambda}} \le SA\left(1 + \sqrt{\frac{1+\lambda}{\lambda}}\right) =: B.$$

For claim (iv), we have

$$L_{k+1}(d) - L_k(d) \le \sum_{s,a}\frac{\tau}{(k+1)^c}\sqrt{\frac{d(s,a)+\lambda}{\rho_{\mathrm{cov}}(s,a)+\lambda}} \le \frac{SA\tau}{(k+1)^c}\sqrt{\frac{1+\lambda}{\lambda}} =: \delta_k.$$

We have

$$\sum_{i=0}^{k}(1-\eta)^i \delta_{k-i} = \tau SA\sqrt{\frac{1+\lambda}{\lambda}}\sum_{i=0}^{k}\frac{(1-\eta)^i}{(k-i+1)^c}.$$

For example, for $c = 2$, the above sum is bounded by $\sum_{n=1}^{\infty} 1/n^2 = \pi^2/6$. Thus, one can set $\xi := \frac{\pi^2 SA}{6}\sqrt{\frac{1+\lambda}{\lambda}}$. $\qquad\square$

## B  Experimental details

Source code is included in the supplemental material.

### B.1  Bidirectional lock

**Environment.**  For the bidirectional lock environment, one of the locks (randomly chosen) gives a larger reward of 1 and the other lock gives a reward of 0.1. Further details on this environment can be found in the work [3].

**Exploration bonuses.** We consider three exploration bonuses:

- Hoeffding-style bonus is equal to

$$\frac{V_{\max}}{\sqrt{N_k(s,a)}},$$

for every $s \in \mathcal{S}, a \in \mathcal{A}$, where $V_{\max}$ is the maximum possible value in an environment which we set to 1 for bidirectional lock.

- We use a Bernstein-style bonus

$$\sqrt{\frac{\mathrm{Var}_{s' \sim P_k(\cdot|s,a)} V_k(s')}{N_k(s,a)}} + \frac{1}{N_k(s,a)}$$

based on the bonus proposed in [37]. $P_k$ denotes an empirical estimation of transitions $P_k(s'|s,a) = N_k(s,a,s')/N_k(s,a)$, where $N_k(s,a,s')$ is the number of samples on transiting to $s'$ starting from state $s$ and taking action $a$.

- MADE's bonus is set to the following in tabular setting:

$$\frac{1}{\sqrt{N_k(s,a)B_k(s,a)}}.$$

**Algorithms.** Below, we describe details on each tabular algorithm.

- **Value iteration.** We implement discounted value iteration given in [37] with all three bonuses.
- **PPO.** We implement a tabular version of the algorithm in [16], which is based on PPO with bonus. Specifically, the algorithm has the following steps: (1) sampling a new trajectory by running the stochastic policy $\pi_k$, (2) updating the empirical transition estimate $P_k$ and exploration bonus, (3) computing Q-function $Q_k$ of $\pi_k$ over an MDP $M_k$ with empirical transitions $P_k$ and total reward $r_k$ which is a sum of extrinsic reward and exploration bonus, and (4) updating the policy according to $\pi_{k+1}(a|s) \propto \pi_k(a|s) \exp(\alpha_k Q_k(s,a))$, where $\alpha_k = \sqrt{2\log(A)/HK}$ based on Cai et al. [16, Theorem 13.1].
- **Q-learning.** We implement Q-learning with bonus based on the algorithms given by Jin et al. [44].

## B.2 Chain MDP

For the chain MDP described in Section 4.2, we run policy gradient for a tabular softmax policy parameterization $\pi(s|a) = \theta_{s,a}$ with the following RL objectives. Since we use a simplex parameterization, we run *projected* gradient ascent.

- **Vanilla PG.** The vanilla version simply considers the standard RL objective $J(\pi_\theta)$. For the gradient $\nabla_\theta J(\pi_\theta)$, see e.g. Agarwal et al. [2, Equation (32)].
- **PG with relative policy entropy regularization.** We use the objective (with the additive constant dropped) given in Agarwal et al. [2, Equation (12)]:

$$L(\pi_\theta) \coloneqq J(\pi_\theta) + \tau_k \sum_{s,a} \log \pi_\theta(a|s).$$

Here, index $k$ denotes the policy gradient step. This form of regularization is more aggressive than the policy entropy regularized objective discussed next. Partial derivatives of the above objective are simply

$$\frac{\partial L(\pi_\theta)}{\partial \theta_{s,a}} = \frac{\partial J(\pi_\theta)}{\partial \theta_{s,a}} + \tau_k \frac{1}{\theta_{s,a}},$$

where the first term is analogous to the vanilla policy gradient.

- **PG with policy entropy regularization.** Policy entropy regularized objective [100, 67, 70, 62] is

$$L(\pi_\theta) \coloneqq J(\pi_\theta) - \tau_k (1-\gamma)^{-1} \mathbb{E}_{(s,a) \sim d_\rho^{\pi_\theta}(\cdot,\cdot)}[\log \pi_\theta(a|s)].$$

The gradient of the regularizer of the above objective is given in Lemma 1.

- **PG with MADE's regularization.** For MADE, we use the following objective

$$L(\pi_\theta) := J(\pi_\theta) - \tau_k \sum_{s,a} \sqrt{d^\pi(s,a)}.$$

The gradient of MADE's regularizer is computed in Lemma 2.

For all regularized objectives, we set $\tau_k = 0.1/\sqrt{k}$.

## B.3 MiniGrid

We follow RIDE [17] and use the same hyperparameters for all the baselines. For ICM, RND, IMPALA, RIDE, BeBold and MADE, we use the learning rate $10^{-4}$, batch size 32, unroll length 100, RMSProp optimizer with $\epsilon = 0.01$ and momentum 0. For entropy cost hyperparameters, we use 0.0005 for all the baselines except AMIGo. We provide the entropy cost for AMIGo below. We also test different values $\{0.01, 0.02, 0.05, 0.1, 0.5\}$ for the temperature hyperparameter in MADE. The best hyperparameters we found for each method are as follows. For **Bebold**, **RND**, and **MADE** we use intrinsic reward scaling factor of 0.1 for all environments. For **ICM** we use intrinsic reward scaling factor of 0.1 for KeyCorridor environments and 0.5 for the others. Hyperparameters in **RIDE** are exactly the same as **ICM**. For **AMIGo**, we use an entropy cost of 0.0005 for the student agent, and an entropy cost of 0.01 for the teacher agent.

## B.4 DeepMind Control Suite

**Environment.** We use the publicly available environment DeepMind Control Suite [95] without any modification (Figure 9). Following the task design of RE3 [86], we use `Cheetah_Run_Sparse` and `Walker_Run_Sparse`.

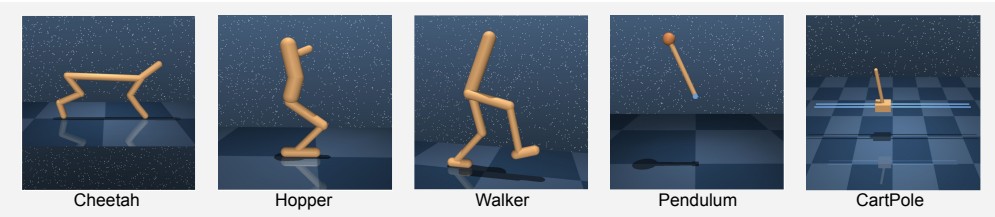

Figure 9: Visualization of various tasks in DeepMind Control Suite. DeepMind Control Suite includes image-based control tasks with physics simulation. We mainly experiment on locomotion tasks in this environment.

**Model-free RL implementations.** For the experiments, we use the baselines of RAD [55], and we conduct a hyperparameter search over certain factors:

- **RND.** We search for the temperature parameter $\tau_0$ over $\{0.001, 0.01, 0.05, 0.1, 0.5, 10.0\}$ and choose the best for each task. Specifically we use $\tau_0 = 0.1$ for `Pendulum_Swingup` and `Cheetah_Run_Sparse`, $\tau_0 = 10$ for `Cartpole_Swingup_Sparse`, and $\tau_0 = 0.05$ for others.
- **ICM.** We search for the temperature parameter $\tau_0$ over $\{0.001, 0.01, 0.05, 0.1, 0.5, 1.0\}$ and choose the best for each task. Specifically we select $\tau_0 = 1.0$ for `Cheetah_Run_Sparse` and $\tau_0 = 0.1$ for the others. For the total loss used in training the networks, to balance the coefficient between forward loss and inverse loss, we follow the convention and use $L_{\text{all}} = 0.2 \cdot L_{\text{forward}} + 0.8 \cdot L_{\text{inverse}}$, where $L_{\text{forward}}$ is the loss of predicting the next state given current state-action pair and $L_{\text{inverse}}$ is the loss for predicting the action given the current state and the next state.
- **RE3**. We use an initial scaling factor $\tau_0 = 0.05$ (the scaling factor of $\tau_k$ at step 0) and decay it afterwards in each step. Note that we use the number of clusters $M = 3$ with a decaying factor on the reward $\rho = \{0.00001, 0.000025\}$. Therefore, the final intrinsic reward scaling factor becomes: $\tau_k = \tau_0 e^{-\rho k}$.
- **MADE.** We search for the temperature parameter $\tau_0$ over $\{0.001, 0.01, 0.05, 0.1, 0.5\}$ and choose the best for each task. Specifically we select $\tau_0 = 0.05$ for `Cartpole_Swingup_Sparse`, `Walker_Run_Sparse` and `Cheetah_Run_Sparse`, $\tau_0 = 0.5$ for `Hopper_Hop` and `Pendulum_Swingup`, and $\tau_0 = 0.001$ for `Quadruped_Run`.

We use the same network architecture for all the algorithms. Specifically, the encoder consists of 4 convolution layers with ReLU activations. There are kernels of size $3 \times 3$ with 32 channels for all layers, and stride 1 except for the first layer which has stride 2. The embedding is then followed by a LayerNorm.

**Model-based Rl implementation**  Here we provide implementation details for the model-based RL experiments. We adopt Dreamer as a baseline and build all the algorithms on top of that.

- **RE3.** For RE3, we follow the hyperparameters given in the original paper. We use an initial scaling factor $\tau_0 = 0.1$ without decaying $\tau_k$ afterwards. The number of clusters is set to $M = 50$. We use a decaying factor on the reward $\rho = 0$.

- **MADE.** We search for the temperature parameter $\tau_0$ over $\{0.0005, 0.01, 0.05, 0.1, 0.5\}$ and choose the best for each map. Specifically we use 0.5 for `Cartpole_Swingup_Sparse`, `Cheetah_Run_Sparse` and `Hopper_Hop`, 0.01 for `Walker_Run_Sparse` and `Pendulum_Swingup` and 0.0005 for `Quadruped_Run`.

## C   Gradient computations

In this section we compute the gradients for policy entropy and MADE regularizers used in the chain MDP experiment. Before presenting the lemmas, we define two other visitation densities. The state visitation density $d^\pi : \mathcal{S} \to [0, 1]$ is defined as

$$d^\pi(s) := (1 - \gamma) \sum_{t=0}^\infty \gamma^t \, \mathbb{P}_t(s_t = s; \pi),$$

where $\mathbb{P}_t(s_t = s; \pi)$ denotes the probability of visiting $s$ at step $t$ starting at $s_0 \sim \rho(\cdot)$ following policy $\pi$. The state-action visitation density starting at $(s', a')$ is denoted by

$$d^\pi_{s',a'}(s, a) := (1 - \gamma) \sum_{t=0}^\infty \gamma^t \, \mathbb{P}_t(s_t = s, a_t = a; \pi, s_0 = s', a_0 = a').$$

The following lemma computes the gradient of policy entropy with respect to policy parameters.

**Lemma 1.**  *For a policy $\pi$ parameterized by $\theta$, the gradient of the policy entropy*

$$R(\pi_\theta) := -\mathbb{E}_{(s,a)\sim d^{\pi_\theta}_\rho(\cdot,\cdot)}[\log \pi_\theta(a|s)],$$

*with respect to $\theta$ is given by*

$$\nabla_\theta R(\pi_\theta) = \mathbb{E}_{(s,a)\sim d^{\pi_\theta}_\rho(\cdot,\cdot)}\left[\nabla_\theta \log \pi(a|s)\left(\frac{1}{1-\gamma}\langle d^\pi_{s,a}, -\log \pi\rangle\right) - \log \pi(a|s)\right].$$

*Proof.*  By chain rule, we write

$$\nabla_\theta R(\pi_\theta) = -\sum_{s,a} \nabla_\theta d^\pi(s,a) \log \pi(a|s) + \sum_{s,a} d^\pi(s,a)\nabla_\theta \log \pi(a|s) = -\sum_{s,a} \nabla_\theta d^\pi(s,a)\log \pi(a|s).$$

The second equation uses the fact that $\mathbb{E}_{x\sim p(\cdot)}[\nabla_\theta \log p(x)] = 0$ for any density $p$ and that $d^\pi(s,a) = d^\pi(s)\pi(a|s)$ as laid out below:

$$\sum_{s,a} d^\pi(s,a)\nabla_\theta \log \pi(a|s) = \sum_s d^\pi(s) \sum_a \pi(a|s)\nabla_\theta \log \pi(a|s) = 0.$$

By another application of chain rule, one can write

$$\nabla_\theta d^\pi(s,a) = \nabla_\theta[d^\pi(s)\pi(a|s)] = \nabla_\theta d^\pi(s)\pi(a|s) + d^\pi(s,a)\nabla_\theta \log \pi(a|s).$$

We further simplify $\nabla_\theta R(\pi_\theta)$ according to

$$\nabla_\theta R(\pi_\theta) = -\sum_{s,a} \nabla_\theta d^\pi(s,a)\log \pi(a|s)$$

$$= -\sum_{s,a} \nabla_\theta d^\pi(s)\pi(a|s)\log \pi(a|s) - \sum_{s,a} d^\pi(s,a)\nabla_\theta \log \pi(a|s)\log \pi(a|s).$$

We substitute $\nabla_\theta d^\pi(s)$ based on Zhang et al. [109, Lemma D.1]:

$$
\begin{aligned}
\nabla_\theta R(\pi_\theta) = &- \frac{1}{1-\gamma} \sum_{s',a'} d^\pi(s',a') \nabla_\theta \log(a'|s') \sum_{s,a} d_{s',a'}(s,a) \log \pi(a|s) \\
&- \sum_{s,a} d^\pi(s,a) \nabla_\theta \log \pi(a|s) \log \pi(a|s) \\
= &\, \mathbb{E}_{(s,a) \sim d_\rho^{\pi_\theta}(\cdot,\cdot)} \left[ \nabla_\theta \log \pi(a|s) \left( \frac{1}{1-\gamma} \langle d_{s,a}^\pi, -\log \pi \rangle \right) - \log \pi(a|s) \right],
\end{aligned}
$$

where $\langle d_{s,a}^\pi, -\log \pi \rangle$ denotes the inner product between vectors $d_{s,a}^\pi$ and $-\log \pi$. This completes the proof. $\qquad\square$

The following lemma computes the gradient of MADE regularizer with respect to policy parameters.

**Lemma 2.** *For a policy $\pi$ parameterized by $\theta$, the gradient of the regularizer*

$$
R(\pi_\theta) := \sum_{s,a} \sqrt{d^\pi(s,a)},
$$

*with respect to $\theta$ is given by*

$$
\nabla_\theta R(\pi_\theta) = \frac{1}{2} \mathbb{E}_{(s,a) \sim d^\pi(\cdot,\cdot)} \left[ \nabla_\theta \log \pi(a|s) \left( \frac{1}{1-\gamma} \langle d_{s,a}^\pi, \frac{1}{\sqrt{d^\pi}} \rangle + \frac{1}{\sqrt{d^\pi(s,a)}} \right) \right].
$$

*Proof.* The proof is similar to that of Zhang et al. [109, Lemma D.3]. We write $\nabla_\theta d^\pi(s,a) = \nabla_\theta d^\pi(s)\pi(a|s) + d^\pi(s,a)\nabla_\theta \log \pi(a|s)$ and conclude based on Zhang et al. [109, Lemma D.1] that

$$
\begin{aligned}
\nabla_\theta R(\pi_\theta) = &\, \frac{1}{2} \sum_{s,a} \frac{\nabla_\theta d^\pi(s,a)}{\sqrt{d^\pi(s,a)}} \\
= &\, \frac{1}{2(1-\gamma)} \sum_{s',a'} d^\pi(s',a') \nabla_\theta \log \pi(a'|s') \sum_{s,a} d_{s',a'}^\pi(s,a) \frac{1}{\sqrt{d^\pi(s,a)}} \\
&+ \frac{1}{2} \sum_{s,a} d^\pi(s,a) \nabla_\theta \log \pi(a|s) \frac{1}{\sqrt{d^\pi(s,a)}} \\
= &\, \frac{1}{2} \mathbb{E}_{(s,a) \sim d^\pi(\cdot,\cdot)} \left[ \nabla_\theta \log \pi(a|s) \left( \frac{1}{1-\gamma} \langle d_{s,a}^\pi, \frac{1}{\sqrt{d^\pi}} \rangle + \frac{1}{\sqrt{d^\pi(s,a)}} \right) \right].
\end{aligned}
$$

$\qquad\square$