# OpenReview forum: "MADE: Exploration via Maximizing Deviation from Explored Regions"
_NeurIPS.cc/2021/Conference — NeurIPS 2021 Poster_

### Official Review · Reviewer_dMyV · 2021-07-12

**Rating:** 5
**Confidence:** 5

**Summary:**

The authors propose adding a regularisation term, that encourages a policy to visit states, less visited by previous policies, to the RL problem to address the exploration-exploitation tradeoff. The result is a convex regularised optimisation problem in the state occupancy of the policy. In addition, the authors propose an algorithm to solve this problem that is based on the Frank-Wolfe algorithm, similar to Hazan et al. i.e., they iteratively find a policy that has a state occupancy with maximal correlation with the negative gradient of the objective. Experiments are performed in several domains and include a comparison with various baselines from the reward free literature.

**Limitations And Societal Impact:**

Not relevant.

**Main Review:**

The problem of balancing exploration and exploitation with function approximation is perhaps one of the most important problems in current DRL research. The authors propose to tackle the problem by introducing a convex regulariser on the state occupancy that will encourage the policy to visit states that were unvisited by previous policies.

However, this construction has two fundamental flaws. The first one is that there are no guarantees that adding a convex regulariser indeed balances the exploration-exploitation tradeoff in an efficient manner. This statement is true in general, so far, no paper in the literature of reward free exploration have claimed to balance exploration and exploitation efficiently. Instead, adding a convex regulariser results in a different problem, that perhaps covers the state space better but does not lead, for example, for an efficient regret algorithm for the RL problem unless it is combined with a proper exploration method. The second issue is that in order to solve the convex regularised problem efficiently, in terms of regret, efficient exploration must be used, on top of the intrinsic reward. These two comments stress that the motivation abstract and intro of the paper are misleading. That said, the proposed algorithm and the convex regulariser are interesting contributions by themselves, taking ideas from the reward free literature as regularisers to the RL setup, and I will continue the review addressing them as a separate problem.

Baselines and related work. The proposed algorithm solves an MDP with an objective that is convex in the state occupancy of the agent, and specifically, a Frank-Wolfe algorithm. That said, I feel that the paper fails to cite relevant papers on the topic. The one paper that is cited (from Hazan et al.) is not compared to empirically, while being the most similar. That said, the baselines that were implemented and compared to are strong baselines and the comparison is done well, I enjoyed reading the experimental section and I feel that if the paper was focused on it, I might have given it a higher score. I believe that further discussion on how to think of these baselines in the lens of convex regularised MDPs would make their introduction better, i.e., do they solve the same problem? I included several papers on Franke Wolfe, FW algorithms in RL and convex regularised MDPs at the end of this review.

Regarding Theorem 1. I feel that the way the theorem was introduced in the main paper is highly misleading. The Theorem states a linear convergence rate for a FW algorithm with an approximate best response which is not a common result. For that reason I checked the supplementary which adds 4 assumptions on the function f in order for this result to hold, with assumption 4 being non trivial in my opinion. The assumption states that there exists a constant that bounds the iteration error of the FW method and the final sample complexity is given in terms of this constant. On the other hand, the authors state that the constants only depend on the MDP parameters. More broadly, the analysis of FW methods with approximate best response is summarised in "Revisiting Frank-Wolfe: Projection-Free Sparse Convex Optimisation", which shows that when an approximate best response is used (the planning oracle that finds the optimal policy in this case) the convergence of FW has a slower rate. The linear convergence in the non approximate case is discussed in "On the Global Linear Convergence of Frank-Wolfe Optimisation Variants" (with MDPs being a special case where d_\pi is in a polytope), but there is not a general result that shows a linear rate with an approximate best response to the best of my knowledge. Furthermore, the linear rate is typically shown for a strongly convex function, which was not assumed by the authors. If the authors can convince me that assumption number 4 is not "stating the linear rate" by construction or will show me some evidence that a similar assumption was used in FW analysis elsewhere, I will consider changing my score. In particular, I would like to see how $\delta_k, \zeta$ and $\tau$ can be expressed as a function of the MDP parameters, without $k$, and without making the strong convexity assumption.

Question: both the \pi_\mix and \rho_cover are mixed policies that weight the previous policies seen by the agent. Therefore, it is not clear what role does the different weights play. Can you elaborate on this?

[1] "Apprenticeship Learning via Inverse Reinforcement Learning"
The first FW algorithm https://ai.stanford.edu/~ang/papers/icml04-apprentice.pdf (see the supplementary)
[2] "Apprenticeship Learning via Frank-Wolfe" on linear convergence for FW in RL https://arxiv.org/pdf/1911.01679.pdf
[3] "Revisiting Frank-Wolfe: Projection-Free Sparse Convex Optimization" FW with approximate best response http://proceedings.mlr.press/v28/jaggi13-supp.pdf
[4] "On the Global Linear Convergence of Frank-Wolfe Optimization Variants" https://arxiv.org/pdf/1511.05932.pdf
[5] "Online Apprenticeship Learning" on why exploration is still needed in the convex regularised problem https://arxiv.org/pdf/2102.06924.pdf
3 papers on solutions to the convex regularised problem:
[6] "Concave Utility Reinforcement Learning: the Mean-field Game viewpoint "https://arxiv.org/pdf/2106.03787.pdf"
[7] "Reward is enough for convex MDPs" https://arxiv.org/pdf/2106.00661.pdf
[8] "Variational Policy Gradient Method for Reinforcement Learning with General Utilities" https://arxiv.org/pdf/2007.02151.pdf

**Time Spent Reviewing:**

3 hours

---

> ### Author Response · Authors · 2021-08-10
> **Response (Part 1/2)**
>
> We thank the reviewer for the time spent reviewing our work, their thorough review, and valuable feedback. We have provided our response to the reviewer's comments and questions below.
>
>
> ### Immediate clarification of our goals in this work
>
> **Sparse reward exploration.** In this paper, we are concerned with conducting exploration in the high-dimensional sparse reward setting. We would like to emphasize that our ultimate goal is to find a policy that *maximizes the standard RL objective*, i.e. the cumulative sum of rewards. Our setting is different from the following problems:
>
> 1. Reward-free (or pure exploration): Here, the agent's goal is to conduct exploration to solve the RL objective for any reward function (Jin et al., 2020).
> 2. Convex MDPs: In convex MDPs, the goal is to solve an alternative objective that is often convex in the policy occupancy density.
> 3. Apprenticeship learning: Here, the goal is to find an optimal policy from expert demonstrations only. In this setting, no reward information is available.
>
> **Different from the convex MDP setting.** We encourage exploration by adding a regularizer to the standard RL objective *that decays over the iterations.* The regularizer decay results in ultimately solving the standard RL objective itself and thus makes it different from the convex MDP setting, such as the one considered in Zahavy et al., 2021.
>
> **A framework that admits the popular count-based methods.** Our framework of adding a regularizer that is a function of visitation density to the RL objective, admits several popular exploration methods including count-based methods. Among these, the Bernstein-type bonus is proved to balance exploration and exploitation near-optimally.
>
> **Simple modification to count-based bonuses.** Our goal is to design a simple and practical algorithm that can be incorporated with deep RL. We show that our particular regularizer results in an intrinsic reward (bonus) that applies a modification to the count-based bonuses, which are proved to have a regret guarantee in the tabular setting. Our intrinsic reward is practically useful as it tackles the online exploration problem by modifying widely-used count-based bonuses used in deep RL algorithms such as RND. In the tabular setting, the Bernstein bonus is shown to capture environment structure (Zannette and Brunskill, 2019). Our tabular experiments show that MADE acts similarly to the Bernstein bonus, suggesting that MADE may also capture certain structural elements.
>
> **Strong empirical performance.** We have empirically demonstrated that MADE is competitive to the Bernstein bonus and works better than the Hoeffding bonus in the tabular setting. This intrinsic reward also achieves state-of-the-art performance in deep RL tasks.
>
>
> ### Guarantees in exploration-exploitation tradeoff via convex regularization
>
> > there are no guarantees that adding a convex regulariser indeed balances the exploration-exploitation tradeoff in an efficient manner.'
>
> We do not claim to have a theoretical guarantee on exploration-exploitation tradeoff, for instance by proving a bound on regret. Instead, efficient exploration by our approach is supported by extensive empirical studies.
>
> As we explain in lines 148 to 154, some popular exploration methods can be viewed as adding a regularizer to the standard objective. In the tabular setting, count-based bonuses are proved to efficiently balance exploration and exploitation. For instance, Hoeffding-type bonus can be expressed by adding a regularizer of the form $\sum_{s,a} \frac{d^\pi(s,a)}{\sqrt{N(s,a)}}$, which is a linear function in $d^\pi(s,a)$ and depends on the visitation of prior policies.
>
> >This statement is true in general, so far, no paper in the literature of reward free exploration have claimed to balance exploration and exploitation efficiently.
>
> We would like to clarify that we consider the sparse reward setting and not the reward-free setting. In the reward-free setting, Zhang et al., 2021 propose a UCB-style algorithm that provably achieves the minimax optimal regret (up to logarithmic factors) in the reward-free setting.
>
> ### Exploration strategy in addition to the regularizer/intrinsic reward
>
> >Instead, adding a convex regulariser results in a different problem, that perhaps covers the state space better but does not lead, for example, for an efficient regret algorithm for the RL problem unless it is combined with a proper exploration method....in order to solve the convex regularised problem efficiently, in terms of regret, efficient exploration must be used, on top of the intrinsic reward.
>
> We agree with the reviewer that one might design an exploration strategy on top of an intrinsic reward. In the paper, we compute an intrinsic reward by taking the gradient of the regularizer. Our exploration method is essentially greedy (taking max or softmax) with respect to an ``optimistic'' Q-function that captures the cumulative sum of *both* extrinsic and intrinsic rewards.
>
> This is indeed one of the common exploration strategies in RL literature, both theory and practice (Jin et al., 2018; Zhang et al. 2021; Jin et al., 2020; Burda et al., 2018). For example, value-based UCB methods add a bonus (such as Hoeffding, Bernstein, or quadratic bonus in linear MDPs) to the extrinsic reward, compute the corresponding Q-function, and then take the greedy action, i.e. $\arg \max_a Q(s,a)$ (Jin et al. 2018, Jin et al. 2020, Zhang et al, 2021).
>
> ### Differences with the reward-free objectives
>
> In reward-free literature, objectives are developed based on state (or state-action) visitation density entropy or KL divergence (Hazan et al., 2019, Zhang et al., 2020). We highlight two important differences between our regularizers and those objectives below:
>
> 1. Our regularizer changes over the iterations. In prior works such as Hazan et al., 2019 or Zhang et al., 2020, the objective is fixed across iterations.
> 2. Our regularizer depends on the visitation of prior policies. This is because the role of the regularizer in our setting is to visit states and actions that have not been fully explored in the past. Therefore, the regularizer depends on the history of visitations. In prior works such as Hazan et al., 2019 or Zhang et al., 2020, the objective is *not* affected by which states and actions have been visited. For instance, the goal of solving an entropy objective is to converge to a policy that can visit states and actions rather uniformly (high entropy $d^\pi$). In contrast, the goal of our regularizer is to gather a uniform set of samples across iterations.
>
> ### Comparison with the literature on FW
>
> While we prove that our objective is solvable via a Frank-Wolfe-based algorithm, this particular algorithm is not the main focus of our work. We agree with the reviewer that a further discussion on Frank-Wolfe and convex MDPs literature is beneficial. We thank the reviewer for providing an extensive list of papers on these topics. We will discuss and cite them in the camera-ready version.
>
> > the paper fails to cite relevant papers on the topic. The one paper that is cited (from Hazan et al.) is not compared to empirically, while being the most similar.
>
> We have provided empirical comparisons with some regularized MDPs such as policy entropy and relative entropy regularization in Section 4.2. We did not compare directly with the algorithm in Hazan et al., 2019 because our focus is on exploration in sparse reward MDPs and not the reward-free setting. Instead, we compare with RE3 which is similar to the algorithm of Hazan et al., 2019 and adds state visitation entropy as a regularizer to conduct exploration. See Figures 6 and 8 for a comparison between MADE and RE3.
>
> ### Comparison with apprenticeship learning
>
> While the Frank-Wolfe method has been used in apprenticeship learning (AL), we would like to emphasize that the formulation in AL is very different from ours. Several differences are discussed below.
> 1. **Difference in setting.** The goal of AL is to find an optimal policy given expert demonstrations without access to a reward function. In our case, however, we have access to extrinsic rewards but not expert demonstrations.
>
> 2. **Difference in objectives.** In AL, the objective is often formulated in a minimax form: finding a policy that performs as well as the expert w.r.t. any reward function. In contrast, our goal is to maximize the expected cumulative reward.
>
> 3. **AL does not involve exploration.** More importantly, online RL is concerned with two aspects: (1) exploration to visit different $(s,a)$ pairs, (2) optimizing an objective. On the contrary, AL is not concerned with exploration and the challenge there is to solve the optimization objective.
>
> 4. **Objective changes over iterations.** In this work, due to using a regularizer, our objective changes over the iterations of the FW method, which is different from AL.
>
>
> ### How the baselines are expressed as convex regularized MDPs
>
> > I believe that further discussion on how to think of these baselines in the lens of convex regularised MDPs would make their introduction better, i.e., do they solve the same problem?
>
> We thank the reviewer for their suggestion. Currently, we have a discussion on how some of the popular baselines (such as visitation-based intrinsic rewards or visitation entropy objectives) can indeed be expressed as regularizers to the RL objective (lines 148 to 154). We will include such relations in the introduction and more discussion to the camera-ready version.

---

> > ### Author Response · Authors · 2021-08-10
> > **Response (Part 2/2)**
> >
> > ### Theorem 1 and the convergence guarantee of Algorithm 1
> >
> > **The four properties in Proposition 1.**
> >
> > >I checked the supplementary which adds 4 assumptions on the function f in order for this result to hold, with assumption 4 being non trivial in my opinion. The assumption states that there exists a constant that bounds the iteration error of the FW method and the final sample complexity is given in terms of this constant. On the other hand, the authors state that the constants only depend on the MDP parameters.
> >
> > We clarify that Theorem 1 provides a guarantee on our particular objective (RL objective plus our regularizer). **In the proof of Theorem 1, we rely on the four *properties* (and not *assumptions*) of our objective** that presented as properties (i) to (iv) in Proposition 1 in the appendix. We emphasize that properties (i) to (iv) are proved in lines 678 to 687 in the appendix for our specific choice of the regularizer. In particular, we have proved that there exists a bound (in terms of the MDP parameters) that bounds the iteration error of the FW method. We provide more details and respond to more specific comments (such as dependency on the MDP parameters) below.
> >
> > Furthermore, note that **property (iv) does not bound the iteration error of the FW method**. Instead, since our objective $L_k(d)$ itself changes over iterations $k$, property (iv) is simply saying that we can design the regularization weight $\tau_k$ so that changes in $L_k(d)$ across iterations are small. Commonly, the objective in FW methods are fixed and do not change across iterations. Thus, the bound in (iv) for FW on a fixed function is zero.
> >
> > Except for the fact that our objective changes across iterations, our proof and guarantee resemble those of Theorem 4.1 in Hazan et al., 2019. That paper has a similar setting to ours, i.e. access to an approximate density and planning oracle, and shows convergence under concavity, boundedness, and smoothness properties.
> >
> > **Convergence of Algorithm 1.**
> >
> > >More broadly, the analysis of FW methods with approximate best response is summarised in "Revisiting Frank-Wolfe: Projection-Free Sparse Convex Optimisation", which shows that when an approximate best response is used (the planning oracle that finds the optimal policy in this case) the convergence of FW has a slower rate. The linear convergence in the non approximate case is discussed in "On the Global Linear Convergence of Frank-Wolfe Optimisation Variants" (with MDPs being a special case where $d_\pi$ is in a polytope), but there is not a general result that shows a linear rate with an approximate best response to the best of my knowledge. Furthermore, the linear rate is typically shown for a strongly convex function, which was not assumed by the authors.'
> >
> > Our guarantee uses an approximate density and planning oracle and the convergence is not linear. We would like to emphasize that
> >
> > 1. *Our proof does not provide a general result for the Frank-Wolfe algorithm with approximate best response* and our result only applies to the particular objective;
> > 2. Our objective is slightly different from the usual objectives in Frank-Wolfe optimization since the objective changes across iterations $k$. Specifically, our regularizer decays over iterations.
> > 3. In this work, we did not focus on making new theoretical contributions. The goal of this result is to show that our objective is solvable, despite being non-concave in the policy.
> >
> > As we prove in Proposition 1, our objective has several properties that aid in proving convergence:
> >
> > 1. Our objective has properties (i) to (iii) in Proposition 1, i.e. it is concave, bounded, and smooth. If such objective is fixed across iterations, Theorem 4.1 of Hazan et al., 2019 proves convergence under these properties.
> > 2. Property (iv) establishes that changes across iterations are small (by appropriately selecting $c$ in the regularizer weights $\tau_k = \tau/k^c$) and do not interfere with the convergence guarantee.
> >
> > >If the authors can convince me that assumption number 4 is not "stating the linear rate" by construction or will show me some evidence that a similar assumption was used in FW analysis elsewhere, I will consider changing my score. In particular, I would like to see how $\delta_k, \xi$ and $\tau$ can be expressed as a function of the MDP parameters, without $k$, and without making the strong convexity assumption.
> >
> > As clarified earlier, number 4 (proved in lines 684 to 687) is a property and not an assumption. $\delta_k, \xi$, and $\tau$ are all also expressed in lines 678 to 687. Specifically, in our results, the following variables are all constants:
> >
> > * $0 <\tau<1$ is the constant of regularization temperature i.e. $\tau_k = \tau/k^c$.
> > * $c$ is also a constant. In line 686, we set $c = 2$.
> > * $\lambda > 0$ is also a constant in the definition of regularizer (e.g. equation (7)).
> >
> > According to the inequality below line 684, $\delta_k$ itself can be bounded by $SA \tau \sqrt{\frac{1+\lambda}{\lambda}}$ using the fact that $c \geq 1$. Since $\lambda$ and $\tau$ are constants, $\delta_k$ is only related to the MDP parameters. Moreover, note that according to the inequalities below line 672, the final sub-optimality is related to a sum over $\delta_k$. As we show in lines 685 to 687, when $c = 2$, this sum is bounded by $\tau \xi$, where $\xi = \frac{\pi^2 S A}{6}\sqrt{\frac{1+\lambda}{\lambda}}$. Since $\lambda$ is a constant, $\xi$ only depends on the MDP parameters.
> >
> >
> > ### Role of different weights in mixture policies
> >
> > > both the $\pi_{\text{mix}}$ and $\rho_{\text{cover}}$ are mixed policies that weight the previous policies seen by the agent. Therefore, it is not clear what role does the different weights play. Can you elaborate on this?
> >
> > We provide a discussion on these terms in lines 123 to 132 and 136 to 139. Our intrinsic reward has the following form $r_i(s,a) = \frac{C}{\sqrt{d^{\pi_{\text{mix}, k}}(s,a) \rho_{\text{cov}}(s,a)}}$.
> >
> > * **Uniform combination of prior policies.** $\rho_{\text{cov}}$ a uniform combination of past visitation densities and characterizes the visitation of $s,a$ pair in previous iterations. In tabular setting, for instance, an empirical estimate is given by $\rho_{\text{cov}} = \frac{N_k(s,a)}{N_k}$, where $N_k$ is the total samples and $N_k(s,a)$ is the visitation count of $(s,a)$ pair at iteration $k$. Thus, our (empirical) intrinsic reward has a term $\frac{1}{\sqrt{N_k(s,a)}}$, which is similar to the count-based bonus (e.g. Hoeffding).
> > * **Geometric combination of policies.** $d^{\pi_{\text{mix}, k}}$ is a combination of past visitation densities via geometric weights. The term $\frac{1}{\sqrt{d^{\pi_{\text{mix}, k}}(s,a)}}$ can be viewed as a correction to the count-only bonus, giving a higher intrinsic reward to the $(s,a)$ pairs visited earlier. Our empirical evaluation suggests that this correction may alleviate some difficulties in the sparse reward setting, namely forgetting and detachment, by encouraging the agent to revisit earlier (``forgotten'') states and actions. In the tabular setting, the Bernstein bonus, which achieves a near minimax optimal rate, also applies a correction (variance of the value function) to the count-only bonus, capturing environment structure (Zanette and Brunskill, 2019). Our tabular experiments show that the new bonus acts similarly to the Bernstein bonus, suggesting that MADE may capture certain structural elements.
> >
> > ### References
> > Zahavy, Tom, et al. "Reward is enough for convex MDPs." arXiv preprint arXiv:2106.00661 (2021).
> >
> > Zanette, Andrea, and Emma Brunskill. "Tighter problem-dependent regret bounds in reinforcement learning without domain knowledge using value function bounds." International Conference on Machine Learning. PMLR, 2019.
> >
> > Zhang, Zihan, Simon Du, and Xiangyang Ji. "Near Optimal Reward-Free Reinforcement Learning." International Conference on Machine Learning. PMLR, 2021.
> >
> > Jin, Chi, et al. "Is Q-Learning Provably Efficient?." Advances in Neural Information Processing Systems 31 (2018): 4863-4873.
> >
> > Zhang, Zihan, Xiangyang Ji, and Simon Du. "Is reinforcement learning more difficult than bandits? a near-optimal algorithm escaping the curse of horizon." Conference on Learning Theory. PMLR, 2021.
> >
> > Jin, Chi, et al. "Provably efficient reinforcement learning with linear function approximation." Conference on Learning Theory. PMLR, 2020.
> >
> > Hazan, Elad, et al. "Provably efficient maximum entropy exploration." International Conference on Machine Learning. PMLR, 2019.
> >
> > Jin, Chi, et al. "Reward-free exploration for reinforcement learning." International Conference on Machine Learning. PMLR, 2020.
> >
> > Burda, Yuri, et al. "Exploration by random network distillation." International Conference on Learning Representations. 2018.
> >
> > ---
> >
> > We thank the reviewer for their suggestions in improving the quality and clarity of the paper. We hope that we have addressed the reviewer’s concerns. We would be happy to answer any questions and discuss further in case the reviewer believes there are any missing details.

---

> > > ### Comment · Reviewer_dMyV · 2021-08-16
> > > **Further questions**
> > >
> > > I would like to thank the authors for posting a detailed response and addressing many of my concerns. I still have a few open questions about this work.
> > >
> > > In their rebuttal the authors stressed that “Different from the convex MDP setting, they introduced a reularizer that decays over the iterations.” However, I found the treatment of the decay schedule in the paper to be lacking and not highlighted enough. While I was reading the rebuttal, I also checked the supplementary material for discussion on the decay schedule and to my surprise, not only that it was not decaying in practice in the DM control suite experiments for MADE, it was also sweeped and picked across a wide array of values (there is a very different optimal value in each domain). I am very confused about \tau_k, when is it k^c and when is it fixed?  Finally, taking inspiration from the exploration-exploitation tradeoff, there is an explicit method for how to decay this parameter, but it only works when we know how the uncertainty estimate of the algorithm decays over time. I did not find any idea in this paper on how to resolve this issue in a DRL setup when such accurate uncertainty estimates are not available. Thus, there is clearly a gap between the popular exploration methods the authors mention and the proposed method, and I don’t understand if the authors make a contribution towards this problem.
> > >
> > > In “Differences with the reward-free objectives” bullet 2,  the authors state: “Our regularizer depends on the visitation of prior policies. This is because the role of the regularizer in our setting is to visit states and actions that have not been fully explored in the past. Therefore, the regularizer depends on the history of visitations. In prior works such as Hazan et al., 2019 or Zhang et al., 2020, the objective is not affected by which states and actions have been visited". This seems to be a misunderstanding by the authors. While the objective in convex MDPs is presented as a function of the state occupancy, FW algorithms use a reward that is the gradient of the convex objective, evaluated on the state occupancy of the mixed policy (i.e., the average of the state occupancies of the previous policies). Thus, the algorithm itself, and the intrinsic reward that is being optimized, is indeed affected by the states visited by previous policies. I request the authors check the Hazan et al. paper and the convex MDP paper to clarify their comment on that front.
> > >
> > > I am still confused about the “Role of different weights in mixture policies”. Concretely, if we consider the finite horizon tabular MDP setup, wouldn’t d_\pi^\mix and \rho_cover be unbiased estimators of the same quantity? The rational here is that the counts of a single trajectory are an unbiassed estimate for the state occupancy of the current policy and then the counts over all history are unbiased estimates of the mixed policy. If not, please state what each of these quantities would be in this case. Secondly, it seems that both d_\pi^\mix and \rho_cover are functions of \pi, thus, when the objective is being differentiated \rho_cover should have a gradient term as well. However, it seems to be considered a fixed quantity during differentiation which doesn’t seem to be correct.
> > >
> > > Regarding the proof in the supplementary, the authors stated that “​​In the proof of Theorem 1, we rely on the four properties (and not assumptions) of our objective” however, in line 672 in the the proof they stated: “By assumption (iv), we write” which is the source of my confusion. This is a small typo so please fix it. Reading the proof of proposition 1, I am a bit confused. The authors state that the diagonal elements are bounded between -¼\lambda^2 to ¼\lambda^2 , and then claim that the hessian is negative definite. This seems to be correct since the objective is a concave function but it is not clear how the authors arrived at that conclusion. It seems to me that the MADE objective is very close to being the norm of the state occupancy, if this is correct it would make the proof much simpler since concavity, smoothness etc are known facts.
> > >
> > > Regarding point 4, for which I was confused about, the derivation of the proof is not very detailed and it is hard for me to verify. I am more interested in understanding the general message of the result. Do you get linear convergence because the regularizer is strongly convex? Or is it because it is convex and you add the polynomial decay? Please help me to understand this part.

---

> > > > ### Author Response · Authors · 2021-08-17
> > > > **Response (Part 1/2)**
> > > >
> > > > We thank the reviewer for their response. We are glad that we have addressed many of the reviewer's concerns. We respond to the reviewer's follow-up questions below.
> > > >
> > > > ### Clarification
> > > >
> > > > We would like to first clarify the following about our work:
> > > > 1. We proposed a framework for exploration by adding regularization to the RL objective. We show that this objective is solvable by a Frank-Wolfe-type algorithm. We do not implement the FW-type algorithm and merely use it to inspire our bonus design.
> > > > 2. We interpret our new bonus as a modified count-based bonus and show that our bonus achieves strong empirical performance.
> > > >
> > > > ### Decay of the regularizer
> > > > > In their rebuttal the authors stressed that “Different from the convex MDP setting, they introduced a reularizer that decays over the iterations.” However, I found the treatment of the decay schedule in the paper to be lacking and not highlighted enough. While I was reading the rebuttal, I also checked the supplementary material for discussion on the decay schedule and to my surprise, not only that it was not decaying in practice in the DM control suite experiments for MADE, it was also sweeped and picked across a wide array of values (there is a very different optimal value in each domain). I am very confused about $\tau_k$, when is it $k^c$ and when is it fixed?
> > > >
> > > > We thank the reviewer for bringing up this point. We realize that we have a typo in Appendix B.4. We wrote sweeping over $\tau_k$, but we meant to write $\tau$ instead, where $\tau_k = \tau/k^c$, where $\tau$ can be viewed as an initial temperature hyperparameter. Indeed in our experiments, the intrinsic reward decays over the iterations. Specifically,
> > > >
> > > > 1. in tabular and MiniGrid experiments, we use $c = 1$;
> > > > 2. in policy gradient experiments, we use $c = 1/2$;
> > > > 3. in DM control experiments, we use $c = 1$ with an additional decay in the form $\exp(-\rho k)$, which results in the final form of $\tau_k = \tau \exp(-\rho k)/k^c$, with $\rho \in${ $0.00001, 0.000025$} as stated in lines 758-759. The additional exponential decay was originally used in RE3 and we also add it to MADE for better empirical performance. Note that even without this decay, our intrinsic reward still decays over iterations with $1/k^c$.
> > > >
> > > > We will highlight our choice of the temperature parameter both in theory and empirical results in the main paper for the camera-ready version.
> > > >
> > > > ### Uncertainty in DRL setup
> > > >
> > > > > Finally, taking inspiration from the exploration-exploitation tradeoff, there is an explicit method for how to decay this parameter, but it only works when we know how the uncertainty estimate of the algorithm decays over time. I did not find any idea in this paper on how to resolve this issue in a DRL setup when such accurate uncertainty estimates are not available. Thus, there is clearly a gap between the popular exploration methods the authors mention and the proposed method, and I don’t understand if the authors make a contribution towards this problem.
> > > >
> > > > Our work does not resolve the challenges in DRL regarding accurate uncertainty estimation. In fact, we are not aware of any implementation of provable uncertainty-based exploration methods for general function approximation that achieve such good empirical performance. Nevertheless, our experiments in the tabular setting show a connection between our bonus and provable Hoeffding and Bernstein bonuses. Both tabular and DRL experiments provide strong evidence that the proposed bonus can encourage exploration.
> > > >
> > > > For clarification of the contributions, this paper does not intend to develop an accurate uncertainty estimation algorithm that provably achieves a good sample complexity. Rather, inspired by the maximizing deviation criterion, we develop an algorithm that: (1) only yields a simple modification of the count-based bonus, (2) has convergence guarantees, and (3) achieves very strong empirical performance in both tabular setting and practical environments.
> > > >
> > > > ### Prior visitations in convex MDPs
> > > >
> > > > > While the objective in convex MDPs is presented as a function of the state occupancy, FW algorithms use a reward that is the gradient of the convex objective, evaluated on the state occupancy of the mixed policy (i.e., the average of the state occupancies of the previous policies). Thus, the algorithm itself, and the intrinsic reward that is being optimized, is indeed affected by the states visited by previous policies. I request the authors check the Hazan et al. paper and the convex MDP paper to clarify their comment on that front.
> > > >
> > > > We agree with the reviewer that the algorithm of Hazan et al., 2019 is also affected by prior visitations. However, that algorithm is designed in such a way that puts *geometric weights on prior state visitations*, due to using a geometric combination of prior policies (i.e. only includes $d^{\pi_{\text{mix}},k}$). In contrast, our approach additionally includes a uniform combination of prior policies, which translates to considering *the (unweighted) total state-action visitations* (i.e. we also include $\rho^k_\text{cov}$). We further discuss this point below.
> > > >
> > > > ### Role of different weights in mixture policies
> > > >
> > > > > I am still confused about the “Role of different weights in mixture policies”. Concretely, if we consider the finite horizon tabular MDP setup, wouldn’t $d_\pi^\text{mix}$ and $\rho_\text{cover}$ be unbiased estimators of the same quantity? The rational here is that the counts of a single trajectory are an unbiassed estimate for the state occupancy of the current policy and then the counts over all history are unbiased estimates of the mixed policy. If not, please state what each of these quantities would be in this case.
> > > >
> > > > No, $d^{\pi_{\text{mix}},k}$ and $\rho^k_\text{cov}$ are not unbiased estimates of the same quantity. In population, these two densities are defines as follows:
> > > >
> > > > 1.  $\rho^k_\text{cov}$ is the density of a **uniform** mixture of prior policies. In particular, $\rho^k_\text{cov}$ is the density corresponding to the policy mixture with policy sequence $(\pi_1, \dots, \pi_k)$ and weights $(\frac{1}{k}, \dots, \frac{1}{k})$. Based on the definition of policy mixture (lines 87-91), density $\rho^k_\text{cov}$ can be expressed as follows (as stated in lines 114-115): $\rho^k_\text{cov}(s,a) = \sum_{i=1}^k \frac{1}{k} d^{\pi_i}(s,a).$
> > > >
> > > > 2. $d^{\pi_{\text{mix}},k}$ is the density of a **weighted** mixture of prior policies. In particular, $d^{\pi_{\text{mix}},k}$ is the density corresponding to the policy mixture with policy sequence $(\pi_1, \dots, \pi_k)$ and weights $((1-\eta)^{k-1}, (1-\eta)^{k-2} \eta, \dots, (1-\eta)\eta, \eta)$ (we note a small typo in line 119), where $\eta$ is the learning rate in line 7 of Algorithm 1. Density $d^{\pi_{\text{mix}},k}$ can be expressed as follows: $ d^{\pi_{\text{mix}},k}(s,a) = (1-\eta)^{k-1} d^{\pi_1}(s,a)+ \sum_{i=2}^k (1-\eta)^{k-i} \eta d^{\pi_i}(s,a). $
> > > >
> > > > Empirically, these densities are estimated. In a tabular finite horizon case with $\gamma = 1$, we can use the following empirical estimation $\hat{d}^{\pi_i}(s,a) = \frac{n_i(s,a)}{n_i}$, where $n_i(s,a)$ and $n_i$ are respectively the visitation count of $(s,a)$ pair and total visitation count $at$ iteration $i$. For simplicity, assume that we take a fixed number of samples in each iteration, i.e. $n_1 = n_2 = \dots = n$ and denote by $N_k = kn$, the total number of samples $by$ iteration $k$. Then,
> > > >
> > > > 1. $\hat{\rho}^k_\text{cov}$ is given by $\hat{\rho}^k_\text{cov}(s,a) = \frac{1}{N_k} \sum_{i=1}^k n_i(s,a) = \frac{N_k(s,a)}{N_k}.$
> > > >     Here $N_k(s,a)$ is the total number of $(s,a)$ pair visitations *by* iteration $k$.
> > > > 2. $\hat{d}^{\pi_{\text{mix}},k}$ is given by $\hat{d}^{\pi_{\text{mix}},k}(s,a) =  (1-\eta)^{k-1} \frac{n_1(s,a)}{n}+ \sum_{i=2}^k (1-\eta)^{k-i} \eta \frac{n_i(s,a)}{n}$.
> > > >
> > > > In summary, the uniform weight density is related to the total prior visitations and we view the other term based on $\hat{d}^{\pi_{\text{mix}},k}$ as a correction.
> > > >
> > > > >Secondly, it seems that both $d_\pi^\text{mix}$ and $\rho_\text{cover}$ are functions of $\pi$, thus, when the objective is being differentiated $\rho_\text{cover}$ should have a gradient term as well. However, it seems to be considered a fixed quantity during differentiation which doesn’t seem to be correct.
> > > >
> > > > Actually, $\rho_{\text{cov}}$ (as defined in lines 114-115) is not a function of $\pi$. It is instead a function of visitations of prior policies $\{\pi_1, \dots, \pi_k\}$. Note that the regularizer $R(d^\pi ; d^{\pi_1}, \dots, d^{\pi_k})$ (defined in (3)) admits visitation densities of policies in prior iterations, i.e. $d^{\pi_1}, \dots, d^{\pi_k}$, as *parameters* (note that we aim at selecting policy $\pi_{k+1}$ given prior visitations). In (4), we simply define $\rho_{\text{cov}}$ to be the average of the prior visitation densities, which acts as a sufficient statistics.

---

> > > > > ### Author Response · Authors · 2021-08-17
> > > > > **Response (Part 2/2)**
> > > > >
> > > > > ### On the proof of Theorem 1
> > > > >
> > > > > > Regarding the proof in the supplementary, the authors stated that “In the proof of Theorem 1, we rely on the four properties (and not assumptions) of our objective” however, in line 672 in the the proof they stated: “By assumption (iv), we write” which is the source of my confusion. This is a small typo so please fix it.
> > > > >
> > > > > We thank the reviewer for pointing out the typo. We will fix it in the camera-ready version.
> > > > >
> > > > > > Reading the proof of proposition 1, I am a bit confused. The authors state that the diagonal elements are bounded between $-1/4\lambda^2$ to $1/4\lambda^2$ , and then claim that the hessian is negative definite. This seems to be correct since the objective is a concave function but it is not clear how the authors arrived at that conclusion. It seems to me that the MADE objective is very close to being the norm of the state occupancy, if this is correct it would make the proof much simpler since concavity, smoothness etc are known facts.
> > > > >
> > > > > As computed below line 679, the $(s,a)$ diagonal element of $\nabla^2 L_k(d)$  is given by
> > > > >
> > > > > $(\nabla^2 L_k(d))_{s,a} = \frac{-\tau}{4 k^c} \times \frac{1}{(d(s,a)+\lambda)^{3/2} (\rho_\text{cov}(s,a) +\lambda)^{1/2}}.$
> > > > >
> > > > > Since $0 < \tau < 1$, the diagonal elements are negative and thus the Hessian is negative definite. Thus, the diagonal elements are indeed between $-\frac{1}{4\lambda^2}$ and zero. In line 680, to prove property (ii), i.e. $-\beta I \preceq \nabla^2 L_k(d) \preceq \beta I$, we set $\beta = \frac{1}{4\lambda^2}$.
> > > > >
> > > > >
> > > > > >  Regarding point 4, for which I was confused about, the derivation of the proof is not very detailed and it is hard for me to verify. I am more interested in understanding the general message of the result. Do you get linear convergence because the regularizer is strongly convex? Or is it because it is convex and you add the polynomial decay? Please help me to understand this part.
> > > > >
> > > > > Our convergence guarantee is indeed *not linear*. While our claims and proofs are correct in the paper (we only claim to prove convergence and not linear convergence in Theorem 1), we mistakenly wrote linear convergence in our previous response instead of just convergence.
> > > > >
> > > > > Note that in Theorem 1, we give the iteration number $K \geq \eta^{-1} \log(10B \epsilon^{-1})$ in terms of $\eta$, which also depends on $\epsilon$ and should satisfy $\eta \leq 0.1 \epsilon \beta^{-1}$ as given in line 673, implying that the convergence is not linear. We realize that not writing $\eta$ explicitly in the statement of Theorem 1 has been the source of confusion and we will fix this in the camera-ready version. The convergence guarantee follows from the concavity, boundedness, and smoothness of the objectives as well as proper decay of the regularizer.
> > > > >
> > > > > ---
> > > > >
> > > > > We thank the reviewer again for their response, thorough review, and follow-up questions. We hope that we have answered the reviewer's questions. We would be happy to answer any other questions or concerns the reviewer might have.

---

> > > > > > ### Comment · Reviewer_dMyV · 2021-08-23
> > > > > > **Response to author's response**
> > > > > >
> > > > > > I thank the authors for the additional details.
> > > > > >
> > > > > > Regarding Decay of the regularizer: this clears a lot of things for me. Please make sure to be extra clear about this.
> > > > > >
> > > > > > The statement in Theorem was confusing without the clarification about $\eta$, please make sure to state the dependence in $\epsilon$ explicitly, this is the most important aspect in the analysis of convex optimisation algorithms. Now, after I understand this part, I would suggest to center the analysis around the fact that the convex part is decaying, this is the main novelty here in terms of analysis, while the rest is standard FW (e.g., following Hazan et al.). I believe that clarifying this would help the reader to understand the novel parts better.
> > > > > >
> > > > > > I am less convinced about the differences between \rho_cover and \pi_\mix. They are both functions of the previous policies that the algorithm have seen, what is the reasoning behind weighting them differently? its ok if this is something that works well in practice, but I am worried about how this different weighing propagates throughout the analysis and the presentation of the objective. It is also not clear why we need two different quantities  \rho_cover and \pi_\mix. \pi_\mix emerges in FW analysis from optimization principles, how would you motivate \rho_cover? to be more clear here, both \pi_mix and \rho_cover have a dynamics aspect in them, i.e., they affect what the reward is given the previous iterations of the algorithm. But \pi_mix stems out from the FW analysis and leads to minimising the convex objective. I am less clear about \rho_cover.
> > > > > >
> > > > > > Now, since the gradient of the objective is evaluated at \pi_\mix and used as a reward, assuming for a minute that \pi_mix=\rho_cover we will get that the objective whose reward you are optimizing is different than the objective you state (but the reward itself won't change). How would that objective look like and will it make sense from the exploration point of view?

---

> > > > > > > ### Author Response · Authors · 2021-08-24
> > > > > > > **Response**
> > > > > > >
> > > > > > > We thank the reviewer for their response and the time spent reviewing our work. We agree with the reviewer regarding clarifying $\eta$ in Theorem 1 and emphasizing the decay of the regularizer in our analysis. We will modify the paper to reflect these suggestions. We respond to the reviewer's questions regarding the differences between $\rho_{\text{cov}}$ and $d^{\pi_{\text{mix}}}$ below.
> > > > > > >
> > > > > > > ### Differences between $\rho_{\text{cov}}$ and $d^{\pi_{\text{mix}}}$
> > > > > > >
> > > > > > > > I am less convinced about the differences between $\rho_{\text{cover}}$ and $\pi_\text{mix}$. They are both functions of the previous policies that the algorithm have seen, what is the reasoning behind weighting them differently? its ok if this is something that works well in practice, but I am worried about how this different weighing propagates throughout the analysis and the presentation of the objective. It is also not clear why we need two different quantities  $\rho_{\text{cover}}$ and $\pi_\text{mix}$. $\pi_\text{mix}$ emerges in FW analysis from optimization principles, how would you motivate $\rho_{\text{cover}}$? to be more clear here, both $\pi_\text{mix}$ and $\rho_{\text{cover}}$ have a dynamics aspect in them, i.e., they affect what the reward is given the previous iterations of the algorithm. But $\pi_\text{mix}$ stems out from the FW analysis and leads to minimising the convex objective. I am less clear about $\rho_{\text{cover}}$.
> > > > > > >
> > > > > > > We present the steps in designing our exploration strategy to clarify how the different weighing of $\rho_{\text{cov}}$ and $d^{\pi_{\text{mix},k}}$ affect the objective and analysis.
> > > > > > >
> > > > > > > **1. Regularizer and its parameters.** To encourage exploration, we propose adding a regularizer $\tau_k R(d^\pi; d^{\pi_1}, \dots, d^{\pi_k})$ to the standard RL objective. We choose the regularizer to be parameterized by visitations of prior policies, i.e. $d^{\pi_1}, \dots, d^{\pi_k}$, because we want to take into account the environment regions visited by prior policies.
> > > > > > >
> > > > > > > **2. Maximizing deviation from explored regions.** We choose a particular form for this regularizer:
> > > > > > >
> > > > > > > $\tau_k R(d^\pi; d^{\pi_1}, \dots, d^{\pi_k}) := \tau_k \sum_{s,a} \sqrt{\frac{d^\pi(s,a)}{\rho_{\text{cov}}(s,a)}}$
> > > > > > >
> > > > > > > We choose to parameterize the regularizer by uniformly combining prior visitations $\rho_{\text{cov}}(s,a) = \sum_{i=1}^k \frac{1}{k} d^{\pi_i}(s,a)$, which \textit{captures the total visitation of $(s,a)$ by prior policies}. For example in the tabular case, the empirical estimate becomes $\hat\rho_{\text{cov}}(s,a) = \frac{ N_k(s,a)}{N_k}$ and therefore $\hat{\rho}_{\text{cov}}(s,a)$ simply captures the total visitation count of $(s,a)$. The total visitation count is a key quantity considered in prior exploration methods such as UCB and appears in concentration bounds such as Hoeffding and Bernstein inequalities.
> > > > > > >
> > > > > > > As we explain in the paper, the reason behind the particular form $\sum_{s,a} \sqrt{\frac{d^\pi(s,a)}{\rho_{\text{cov}}(s,a)}}$ is that later when we take the gradient to construct our exploration bonus, the bonus resembles existing count-based bonuses multiplied by a correction term. This makes the approach practical as it simply adjusts the existing count-based methods.
> > > > > > >
> > > > > > > **3. Obtaining the bonus using iterations of the FW.** We then use an FW-type algorithm to solve the regularized objective. The reason behind choosing this algorithm is to exploit the concavity of the regularized objective w.r.t. $d^\pi$. The FW method tells us that we can solve this objective as follows:
> > > > > > >
> > > > > > >   1. Find a solution to $d^{\pi_{k+1}} \in \arg \max_{d^\pi} \langle d^\pi, \nabla_{d^\pi} L_k(d^\pi) \big|_{d^\pi = d^{\pi_\text{mix, k}}} \rangle$.
> > > > > > >   2. Combine $d^{\pi_{k+1}}$ and previous densities via geometric weights.
> > > > > > >
> > > > > > > Notice that step 1 is exactly solving *planning with a reward function* $\nabla_{d^\pi} L_k(d^\pi) \big|_{d^\pi = d^{\pi_\text{mix, k}}}$. Therefore, our reward at iteration $k$ becomes equation (5). As the reviewer states, the FW-type algorithm naturally introduces the geometrically weighted mixture density $d^{\pi_\text{mix,k}}$.
> > > > > > >
> > > > > > > In summary, we choose to *parameterize* the regularize by the uniform mixture $\rho_{\text{cov}}(s,a)$ because all prior visitations are equally important (and this term appears in all count-based exploration methods). The geometric mixture $d^{\pi_{\text{mix},k}}$ is a byproduct of the optimization algorithm.
> > > > > > >
> > > > > > > > Now, since the gradient of the objective is evaluated at $\pi_\text{mix}$ and used as a reward, assuming for a minute that $\pi_\text{mix}$=$\rho_{\text{cover}}$ we will get that the objective whose reward you are optimizing is different than the objective you state (but the reward itself won't change). How would that objective look like and will it make sense from the exploration point of view?
> > > > > > >
> > > > > > > Note that we choose the regularization parameter $\tau_k$ to decay polynomially and thus, as long as the regularizer is bounded (we add a constant to the denominator to ensure that), the objective quickly converges to the standard RL objective.
> > > > > > >
> > > > > > > To answer the reviewer question, suppose that we choose to parameterize the regularizer by a geometric mixture of the prior visitations, i.e. we set $\rho_{\text{cov}} := d^{\pi_{\text{mix},k}}$. Then the objective becomes
> > > > > > >
> > > > > > > $(1-\gamma)^{-1}\sum_{s,a} d^\pi(s,a) r(s,a) + \tau_k \sum_{s,a} \sqrt{\frac{d^\pi(s,a)}{d^{\pi_{\text{mix},k}}(s,a)}}$
> > > > > > >
> > > > > > > As before, we compute the reward at iteration $k$ by taking the gradient of the objective w.r.t. to $d^\pi$ at $d^\pi = d^{\pi_{\text{mix},k}}$:
> > > > > > >
> > > > > > > $r(s,a) + \frac{\tau_k/2}{d^{\pi_{\text{mix},k}}(s,a)}$
> > > > > > >
> > > > > > > So our bonus becomes $\frac{\tau_k/2}{d^{\pi_{\text{mix},k}}(s,a)}$. This new bonus encourages visiting $(s,a)$ pairs with small $d^{\pi_{\text{mix},k}}(s,a)$, i.e. states and actions that are not sufficiently visited by the most recent policies. However, unlike MADE bonus, this new bonus does not have a term that captures the total *unweighted* number of visitations for $(s,a)$ pair.
> > > > > > >
> > > > > > > -----
> > > > > > >
> > > > > > > We hope that we have addressed the reviewer's questions. We would be happy to discuss further or answer any more questions the reviewer might have.

---

> > > > > > > > ### Comment · Reviewer_dMyV · 2021-08-24
> > > > > > > > **Respond to authors**
> > > > > > > >
> > > > > > > > I thank the authors for engaging in the rebuttal period and clarifying certain parts in their paper. I hope that the authors will take these points into account when preparing the next version of this paper. I am much more confident about their work following the rebuttal and I therefore decided to increase my score. That said, I remain uncertain about \rho_cover and \pi_\mix and therefore I don't feel that I can recommend an acceptance. I will therefore increase my score from 4 to 5. Thanks for the great discussion.

---

### Official Review · Reviewer_Xa1c · 2021-07-13

**Rating:** 6
**Confidence:** 3

**Summary:**

The paper proposes an alternative exploration strategy by MAximizing the DEviation (MADE) of the next policy from explored regions of prior policies by adding an adaptive regularizer (function of state-action visitation counts) to the primary RL objective at each iteration to balance between exploration the new state space and exploit the visited states.  In addition, MADE has been theoretically proven to converge to the global optimum given access to an approximate planning oracle.  The authors show that MADE applies some simple adjustments to the Hoeffding-style count-based bonus. Moreover, experiments in three different RL algorithms (Value iteration, PPO, Q-Learning) show the superiority over the state-of-the-art UCB-bonus methods (Hoeffding and Bernstein bonus). In addition, the authors demonstrate that MADE can practically work both in Model-based (Dreamer) and Model-Free (IMPALA, RAD) RL algorithms, significantly improving sample-efficiency over baselines.

**Limitations And Societal Impact:**

Not applicable.

**Main Review:**

The paper proposes an interesting method for addressing the problem of exploration in RL, with theoretical results about the global convergence to the optimum and solid empirical performance compared to the SOTA. Overall, I like the paper, but I still have few doubts that I would like the authors to address:

- It is not clear how we can get the total reward at iteration k by taking the gradient of the new objective as shown in equation (5).
- As far as I understand, can $r(s,a)$ be derived from equation (2)?
- But then still, how is that the total reward can be derived from the gradient of the regularized objective function?

**Time Spent Reviewing:**

5

---

> ### Author Response · Authors · 2021-08-10
> **Response**
>
> We thank the reviewer for the time spent reviewing our work, their thoughtful review, and interest in our work.
>
> The reviewer raises interesting and important questions on (i) how taking the gradient of the objective gives the total reward at iteration $k$ in Algorithm 1, and (ii) can the reward be computed from equation (2). The reason that the objective gradient in each iteration acts as the reward in the planning step is a byproduct of the optimization procedure in Algorithm 1. Below, we elaborate on this point.
>
> Equation (2) is the standard RL objective $J(d^\pi) = (1-\gamma)^{-1} E_{s,a \sim d^\pi(.,.)}[r(s,a)] =(1-\gamma)^{-1} \langle d^\pi, r \rangle$, where $r$ and $d^\pi$ denote vectors with elements $r(s,a)$ and $d^\pi(s,a)$, respectively. Recall that our goal is to compute a policy $\pi^\star \in \arg \max_\pi J(d^\pi)$. To encourage exploration, we propose adding a regularizer $R(d^{\pi}; d^{\pi_{1:k}})$ to this objective, which is a concave function of in the visitation density: $L_k(d^\pi) = J(d^\pi) + \tau_k R(d^\pi; \{d^{\pi_i}\}_{i=1}^k).$ Note that this objective is not concave in $\pi$.
>
> Fortunately, this objective is concave in the visitation density $d^\pi$ and instead, one can solve the following constrained concave optimization problem: $\max_{d^\pi \in \mathcal{K}} L_k(d^\pi),$
> where $d^\pi$ is the set of all valid visitation densities in the MDP. To solve this constrained objective, we use an algorithm based on the conditional gradient method (Frank and Wolfe, 1956). The conditional gradient method tells us that we can solve this objective as follows:
>
> 1. Find an solution to $d^{k+1} \in \arg \max_d \langle d, \nabla_d L_k(d) \big|_{d = d^k}\rangle$.
>  2. Combine $d^{k+1}$ and previous densities via geometric weights.
>
> Comparing step 1 with the standard RL objective, one may notice that this step is exactly solving *planning with a reward function* $\nabla_d L_k(d) \big|_{d = d^k}$. Therefore, our reward at iteration $k$ becomes equation (5). So we do not compute rewards using equation (2).
>
> ### References
> Frank, Marguerite, and Philip Wolfe. "An algorithm for quadratic programming." Naval research logistics quarterly 3.1-2 (1956): 95-110.
>
> ----
>
> We hope that we have addressed the reviewer's questions. We would be happy to answer any more questions and discuss further.

---

### Official Review · Reviewer_2ECg · 2021-07-16

**Rating:** 7
**Confidence:** 4

**Summary:**

The paper proposes an new approach for exploration in reinforcement learning (RL). The idea is to give an exploration bonus to visiting with the new policy state-action pairs which have low visitation density compared to previously visited state-action pairs. The paper provides a convergence proof assuming a perfect oracle provides visitation densities for a chosen mixture policy. The proposed approach is compared in experiments to RL baselines in both continuous and discrete control tasks with function approximation and in tabular benchmarks.

**Limitations And Societal Impact:**

Yes

**Main Review:**

The idea behind the proposed exploration bonus is not surprising by itself. There are other methods that try to explore in parts which have not been explored before. For example, [Hong et al., 2018] adds an exploration bonus for a policy which is different from previous policies. In addition, as mentioned in this paper count based methods also try to visit state-action pairs which have not been visited before. One of the main contributions is to propose an exploration bonus that is efficient and simple and can be implemented in practice.

Authors: how is the proposed approach related to prior work where the new policy is regularized by maximizing the difference to prior policies such as [Hong et al., 2018]? Intuitively, maximizing the difference to prior policies also maximizes the differences to prior state-action distributions indirectly. Is there a benefit to the approach you are using? An experimental comparison to [Hong et al., 2018] would make sense.

The submission appears technically sound. The theoretical proof is nice. It would be good to have a discussion on what the theoretical proof means in practice since there is the assumption on the oracle providing visitation densities. The discussion on how the proposed exploration bonus can be seen as another form of other exploration bonuses is interesting and valuable. The empirical results show the approach outperforming baselines and the results are analyzed in a reasonable way. Having both tabular and continuous tasks is valuable.

The paper should provide timing results. How much computation time does each method use per sample? Is the proposed approach computationally intensive compared to other approaches?

The paper is clear and well written. Emphasizing even more that Algorithm 1 is a theoretical proof but not actually used in the experiments would improve clarity. The practical implementation of the methods is one of the key things for a practitioner in RL and this could be also emphasized (although it is relatively straightforward to find it in the text).

I expect the results of the paper to be used by other researchers. While similar ideas have been proposed before, exploration is an important unsolved problem and the idea proposed in this paper is simple and seems to perform well in the experiments.

[Hong et al., 2018] Hong, Z.W., Shann, T.Y., Su, S.Y., Chang, Y.H., Fu, T.J. and Lee, C.Y., 2018, December. Diversity-driven exploration strategy for deep reinforcement learning. In Proceedings of the 32nd International Conference on Neural Information Processing Systems (pp. 10510-10521).

UPDATE: I am happy with the author's rebuttal and the changes they are planning for the next version of the paper.


**Time Spent Reviewing:**

6

---

> ### Author Response · Authors · 2021-08-10
> **Response**
>
> We thank the reviewer for the time spent reviewing our work, their thorough review, and positive feedback. We have provided our response to the reviewer's comments and questions below.
>
> ### Comparison with maximizing difference to the prior policies
> > how is the proposed approach related to prior work where the new policy is regularized by maximizing the difference to prior policies such as [Hong et al., 2018]? Intuitively, maximizing the difference to prior policies also maximizes the differences to prior state-action distributions indirectly. Is there a benefit to the approach you are using?
>
> This is a very interesting question. Indeed, using a regularizer that is a function of visitation densities and maximizies the difference to prior visitations have several benefits, which we highlight below.
>
> 1. **Exploration goal.** The goal of exploration is to visit diverse state-action pairs and not necessarily to try maximally different policies. Our regularizer encourages the new policy to visit state-action pairs that are not visited by prior policies. Based on this, we now explain why maximizing distance of the next policy to prior policies may not achieve this goal:
>      * **Having significantly different visitation densities does not necessarily imply significantly different policies.** Consider for instance, an MDP with state space $\mathcal{S} = \{s_0\} \cup \mathcal{S}_1 \cup \mathcal{S}_2$, action space $\mathcal{A} = \{a_1, a_2\}$, and initial state $s_0$. Suppose that taking action $a_1$ from state $s_0$ moves the agent to $\mathcal{S}_1$ w.p. 1 and the agent remains in $\mathcal{S}_1$ regardless of its policy. Similarly, taking action $a_2$ from state $s_0$ moves the agent to $\mathcal{S}_2$ w.p. 1 and the agent remains in $\mathcal{S}_2$. Now consider two policies: (1) $\pi_1$ that takes $a_1$ at $s_0$ and takes $a_1$ at all other states, and (2) $\pi_2$ that takes $a_2$ at $s_0$ and takes $a_1$ at all other states. These two policies are the same except at $s_0$ yet, policy $\pi_1$ only visits $\{s_0\} \cup \mathcal{S}_1$ whereas policy $\pi_2$ only visits $\{s_0\} \cup \mathcal{S}_2$ and they have a significantly different visitation densities.
>      * **Having significantly different policies does not necessarily imply significantly different visitation densities.** For this, consider an MDP where actions $a_1$ and $a_2$ both have the exact same dynamics $P(s'|s,a_1) = P(s'|s,a_2)$ (and we are not provided with this information). A policy that takes $a_1$ at every state looks significantly different from a policy that takes $a_2$ from every state yet, these two policies have the same visitation density.
> 2. **Benefits in optimization objective.** The standard RL objective is linear in visitation density but non-concave in the policy. When we add a regularizer that is a function of visitation density and concave, the objective remains concave in visitation density, which allows us to use Algorithm 1, define rewards according to the gradient of the objective (equation (5)), and obtain theoretical guarantee of Theorem 1. Adding a regularizer that is a function of policy may result in a new objective that is not concave in policy nor visitation density.
>
> 3. **Benefits in implementation.** Our approach results in the following practical benefits:
>      * **Simple bonus-based method.** It results in a simple bonus-based exploration strategy.
>      * **Less memory.** Our approach does not require saving prior policies or prior visitation measures. We only require limited samples from the most recent policies. On the other hand, methods based on policy regularization may need to save all/most prior policies.
>
> ### Further comparison with Hong et al., 2018
>
> We thank the reviewer for suggesting the work Hong et al., 2018. We have provided a detailed comparison of our approach to maximizing difference w.r.t. prior policies in the above segment. Here, we further compare our method with Hong et al., 2018 and will include the discussion in the camera-ready version.
>
> **Algorithmic differences**
>
> 1. Hong et al., 2018 proposed to use a KL-Divergence regularized objective for encouraging the agent to take different action w.r.t. the previous policies. In contrast, our algorithm modifies count-based bonuses and maximizes the deviation from previous visitations. Thus, instead of putting constraints on policy directly, we derive an intrinsic reward that puts constraints on the visitation density of the next policy. This approach has several benefits that were described earlier.
> 2. The algorithm in Hong et al., 2018 requires more hyperparameters (e.g. clipping distance measure and adaptive scaling) compared to our method which uses only two hyper-parameters.
>
> **Empirical comparison**
>
> The algorithm in Hong et al., 2018 is based on Q-learning/DDPG while MADE is based on IMPALA/RAD/Dreamer algorithms. Due to the differences in the base RL algorithms, direct empirical comparison of the two methods on all experiments may be unfair. Furthermore, we were not able to find an open-source implementation of Hong et al., 2018. We plan to contact the authors and do a fair comparison with their method for the camera-ready version.
>
> ### Computational and timing results for MADE
> We thank the reviewer for pointing this out. MADE indeed doesn't require much more computation compared to previous methods. We provide some rough timing measures here. For model-free DM Control, RAD+MADE is ~10.5FPS, RAD+RND is ~11.8FPS, RAD+ICM is ~5.88FPS, RAD+RE3 is ~13.7FPS; For model-based DM Control, the training speed for Dreamer+MADE is ~20FPS, Dreamer+RE3 is ~30FPS, Dreamer is ~30FPS. We could see that MADE yields a decent runtime compared with other baselines. Note, however, that optimizing the runtime was not the focus of our paper. There could be other algorithm or system-level optimization methods we can use that enables a shorter runtime, which can be done for practical purposes.
>
> ----
>
> We thank the reviewer for their suggestions in improving the quality and clarity of the paper. Based on the reviewer's suggestions, we will add discussions on implications of Theorem 1, emphasis on practical implementation details, and comparison with Hong et al., 2018. We hope that we have addressed the reviewer’s questions. We would be happy to answer any more questions and discuss further.

---

### Official Review · Reviewer_zDng · 2021-07-16

**Rating:** 7
**Confidence:** 4

**Summary:**

This paper introduces a new exploration technique called MADE that computes a bonus that depends on the ratio between the visitation densities of the current policy and policy cover of previous policies. This incentivizes the agent towards it has not been before and leads to more exploration. The authors provide a convergence result for this bonus and evaluate it on grid worlds, Mini Grid and Mujoco environment to show that it leads to better exploration than existing techniques.

**Limitations And Societal Impact:**

The authors have adequately addressed the limitations and potential negative societal impact of their work

**Main Review:**

Overall I think that this is a good paper, it builds on previous ideas about using policy covers for exploration but is able to derive a bonus that can be used with any reinforcement learning algorithm, model based or model free and not limited to policy gradient method. I found the paper well written and easy to follow. The empirical evaluation was really thorough, MADE was evaluated on grid worlds, discrete actions and continuous control environments.
MADE showed really good results on MiniGrid, however on DM control while results look promising it is hard to interpret them without knowing the number of seeds used for each.

How was Eq (4) chosen vs other possible solutions? Could we use those insights to derive other exploration bonuses? And could we hope to find an optimal regularizer?
In section 4.1 and Figure 3 MADE is shown to work similarly to Bernstein bonuses which are as good as we can hope. Does it mean that this choice of regularizer is somewhat optimal?

L139: What happened to the term $\sqrt{N_k}$? To match equation 4 it should be included and would lead to different bonuses.
I found the paragraph L148 to be interesting, being able to fit any bonus through the MADE paradigm could be powerful, however little to no insight is actually gained from this. Do you think there is anything we can learn through this point of view?

When looking at Equation (4) it is not clear that the bonus actually goes down to zero. Is that the case? If not that might explain the regularizer instead of exploration bonus. In that case can the bonus impact the reward and change the optimal policy of the problem being solved?

Small typos:
L194 "gradients.= In"
L215 "we as the training"




**Time Spent Reviewing:**

4

---

> ### Author Response · Authors · 2021-08-10
> **Response**
>
> We thank the reviewer for the time spent reviewing our work, their thorough review, and their interest in our work. The reviewer brought up several interesting questions to which we respond below.
>
> ### Number of random seeds in the DM Control
> We used 4 random seeds in all of our experiments in DM Control. We will add this detail to the camera-ready version.
>
> ### Designing regularizers for exploration
>
> >How was Eq (4) chosen vs other possible solutions?
>
> We aimed at designing a regularizer that satisfies the following criteria.
> 1. **Easy implementation.** We were looking for a regularizer that gives an intrinsic reward that is simple to implement. This choice particularly interests us because the corresponding intrinsic reward simply modifies the count-based intrinsic reward. So one can just multiply this correction (i.e. $1/\sqrt{d^{\pi_{\text{mix}, k}}}$) to an existing count-based bonus such as RND for a significant improvement in performance.
>
>   2. **Technical properties.** We wanted the regularizer to satisfy certain properties such as concavity so that we can solve the objective (e.g. via Algorithm 1).
> 3. **Visiting unexplored regions.** Intuitively, the proposed objective can be understood as encouraging a policy that maximizes a sum of RL objective (i.e. exploit given current estimation) and a regularizer that maximally focuses on unvisited regions of the MDP.
> 4. **Strong empirical performance.** We were also looking for a regularizer that performs well empirically. In practice, both finite sample properties and optimization properties play important roles, and our experiments, suggesting that this regularizer offers significant benefits on both fronts.
>
> ### Insights from MADE
>
>  In addition to the points discussed above, our empirical evaluation suggests that the correction term $1/\sqrt{d^{\pi_{\text{mix}, k}}}$ may alleviate some difficulties in the sparse reward setting, namely forgetting and detachment, by encouraging the agent to revisit earlier (``forgotten'') states and actions.
>
> > In section 4.1 and Figure 3 MADE is shown to work similarly to Bernstein bonuses which are as good as we can hope. Does it mean that this choice of regularizer is somewhat optimal?
>
> In the tabular setting, the Bernstein bonus, which achieves a near minimax optimal rate, also applies a correction (variance of the value function) to the count-only bonus, capturing certain structures of the environment (Zanette and Brunskill, 2019). Our tabular experiments show that the new bonus acts similarly to the Bernstein bonus, suggesting that it may capture certain structural elements.
>
> Regarding optimality, whether MADE bonus is somewhat optimal requires further theoretical analysis. However, in practical settings, unlike the Bernstein bonus, MADE is easy to compute and can be combined with function approximation like neural networks, whereas Bernstein is hard to compute beyond the tabular settings.
>
> ### Using the insights for other exploration methods
>
> > Could we use those insights to derive other exploration bonuses? And could we hope to find an optimal regularizer?....I found the paragraph L148 to be interesting, being able to fit any bonus through the MADE paradigm could be powerful, however little to no insight is actually gained from this. Do you think there is anything we can learn through this point of view?
>
> Our approach gives several insights and motivates new directions for future exploration methods:
> * As we discuss in the paper, many exploration strategies (e.g. bonus-based methods) can fit into the paradigm of adding a regularizer to the RL objective. Two criteria are important for an algorithm to succeed in practice: (1) an effective exploration strategy so that the empirical objective estimates the true objective well, and (2) an objective that is easy to optimize. The regularized objective may be useful to study both exploration and optimization aspects under the same framework.
> * As explained previously, the similarity of MADE to Bernstein suggests that MADE may exploit problem structure. An interesting future direction is to understand problem structures and exploit them for more efficient exploration.
> * We are not aware of any existing works that try to find an optimal regularizer. It is certainly an interesting direction for future work.
>
>
>
> ### Specific questions
>
> > L139: What happened to the term $\sqrt{N_k}$? To match equation 4 it should be included and would lead to different bonuses.
>
> Yes, $\sqrt{N_k}$ affects the bonus value. In line 139, we use $\propto$ and only write the terms that change over different states and actions. Since $N_k$ only depends on iteration $k$, we can combine it with the temperature parameter $\tau_k$ in equation (3).
>
> > When looking at Equation (4) it is not clear that the bonus actually goes down to zero. Is that the case? If not that might explain the regularizer instead of exploration bonus. In that case can the bonus impact the reward and change the optimal policy of the problem being solved?
>
> While the regularizer in equation (4) does not go to zero, we set the temperature parameter $\tau_k$ in equation (3) so that the regularizer converges to zero as $k$ increases. For the guarantee of Theorem 1, we show that setting $\tau_k = \tau/k^2$ is sufficient; see e.g. lines 686 and 687 in Appendix A. In experiments, as discussed in lines 174 and 175, we use the empirical counts (or pseudo-counts) in place of the densities, resulting in a bonus that automatically goes to zero over the iterations. As the reviewer states, indeed if the bonus does not go to zero then that impacts the reward and changes the optimal policy.
>
> We thank the reviewer for pointing out the typos in the paper. We will address them in the camera-ready version.
>
> ----
>
> We hope that we have addressed the reviewer's questions. We would be happy to answer any more questions and discuss further.

---

### Decision · Program_Chairs · 2021-09-27

**Decision:**

Accept (Poster)

**Comment:**

The initial reviews were overall positive (and raised concerns well addressed in the rebuttal), except for one review that raised concerns about the proof of the provided theoretical result. This led to an engaged discussion between the authors and the reviewers, that clarified many aspects of the proof and of the approach. Some questions remained about the interpretation of some quantities ($\rho_{cover}$, which is part of the iteration-dependent regularizer, and $d^{\pi_{mix}}$, the iterate computed by the algorithm). Maybe that making a parallel with mirror descent could help clarify this aspect (for the idea of regularizing wrt the last iterates, even though $\rho_{cover}$ is not the last iterate due to different weighting of the involved occupancy measures, and even though the regularizer is not a Bregman divergence).

Overall, I do think that this helpful discussion addressed sufficiently the initial concerns, and that the theoretical result is technically correct. Beside, this is not the sole contribution of this paper, it also provides a strong empirical analysis, where the approach compares favourably to strong baselines.

To sum up, I think this paper would be a solid addition to the neurips program, and I recommend its acceptance.

I strongly encourage the authors to revise their paper taking into account the discussion with reviewer dMyV as well as other reviews (notably, it is more than Frank-Wolfe, but less general), I think it will widen the audience and increase the impact of this contribution.